# The Walloon farmers position differently their ideal dairy production system between a global-based intensive and a local-based extensive model of farm

Anne-Catherine Dalcq[1], Thomas Dogot[1], Yves Beckers[1], Yves Brostaux[1], Eric Froidmont[2], Frédéric Vanwindekens[2], Hélène Soyeurt[1] *

1 Gembloux Agro-Bio Tech, TERRA Teaching and Research Centre, University of Liège, Gembloux, Belgium, 2 Walloon Research Centre, Gembloux, Belgium

* hsoyeurt@uliege.be

## Abstract

Dairy farming systems are evolving. This study presents dairy producers' perceptions of their ideal future farm (**IFF**) to ensure revenue, and attempts to determine the reasons for this choice, the environmental aspects related to this choice, the proximity between the current farm and the IFF and the requirements for reaching this IFF. Just before the end of the European milk quota, a total of 245 Walloon dairy producers answered a survey about the characteristics of their IFF and other socio-environmental-economic information. A multiple correspondence analysis (**MCA**) was carried out using seven characteristics of the IFF (intensive *vs.* extensive, specialised *vs.* diversified, strongly *vs.* weakly based on new technologies, managed by a group of managers *vs.* an independent farmer, employed *vs.* familial workforce, local *vs.* global market, standard *vs.* quality-differentiated production) to observe the relationships between them. Based on the main contributors to the second dimension of the MCA, this axis was defined as an IFF gradient between the local-based extensive (**LBE**) producers (26%) and the global-based intensive (**GBI**) producers (46%). The differences of IFF gradient between modalities of categorical variables were estimated using generalised linear models. Pearson correlations were calculated between the scores on the IFF gradient and quantitative variables. Finally, frequencies of IFF characteristics and the corresponding characteristic for the current situation were calculated to determine the percentages of "unhappy" producers. Some reasons for the choice of IFF by the producers have been highlighted in this study. Environmental initiatives were more valued by LBE than GBI producers. Low similarity was observed between the current farm situation of the respondents and their IFF choice. LBE and GBI producers differed significantly regarding domains of formation (technical and bureaucratic *vs.* transformation and diversification respectively) and paths of formation (non-market *vs.* market respectively). Two kinds of farming systems were considered by dairy producers and some socioeconomic and environmental components differed between them.

**Data Availability Statement:** Data were collected by the Committee of 'Carrefour des Productions Animales', a committee of stakeholders of animal

sector in the Walloon Region of Belgium (research center, agricultural unions, university). The data are stored on the CAMI platform of Gembloux Agro-Bio Tech. A request can be made to the President of the platform Mr Gengler (Nicolas. Gengler@uliege.be) to get data.

**Funding:** The author(s) received no specific funding for this work.

**Competing interests:** The authors have declared that no competing interests exist.

## Introduction

The progressive organisation of society during the Neolithic period has led to the appearance of "producers" who are responsible for producing food for more than just themselves and their family [1, 2]. Since World War II, public policies have been set up to increase food production [3]. These policies impacted the development of producers and their farms in the European Union. In the southern part of Belgium, the mean number of cows and the mean agricultural area per producer increased between 1980 and 2017 from 20 to 66 heads and from 25 to 71 hectares, respectively [4].

Producers are now facing great challenges to stay profitable. The price of the inputs (e.g. buildings, agricultural machinery, installations, feeding, veterinary care) of dairy production (**DP**) are increasing while the milk price shows great variability and its inflation is not similar to that observed for the inputs [5, 6]. Moreover, the European Union has decreased financial support to farmers [7]. On 1st April 2015, the European Union removed the quota system which had managed the supply of DP [8]. This led to greater milk price volatility. Additionally, sanitary crises such as mad cow disease (bovine spongiform encephalopathy (BSE)) and the dioxine crisis, among others, have shocked consumers and led to new rules and regulations at European level and to the creation of food security agencies in its countries. Moreover, these episodes modified consumers' behaviours regarding their food purchases, they asked for more transparency and directed themselves towards organic food or local chains [9]. Besides the economic view, the impacts of farming on the environment have been noted and policies have been set up in the Common Agricultural Policy to solve these problems [6, 10].

In this context, the question often asked by dairy producers and stakeholders of the dairy sector is what the future of dairy farming entails, how to remain profitable and more generally sustainable. Several authors, such as Napoléone *et al.* [11], Havet *et al.* [5] and Lebacq [6], have studied the evolution of dairy farming and the present dairy systems, finding trends that exist in the sector. For instance, the project Mouve, funded by the French National Research Agency, studied the evolution of dairy farming systems in 6 dairy basins around the world. Their results gathered the publications of Napoléone *et al.* [11] and Havet *et al.* [5]. Moreover, some other authors (e.g., Bergevoet *et al.* [12], Methorst *et al.* [13], Weltin *et al.* [14] and Verhees *et al.* [15] have studied the future paths of development considered by dairy producers. These studies were performed on the basis of data from 2001 to the beginning of 2013. They explored some reasons for these choices for the future.

This study is innovative as it asks what is the ideal future farm (**IFF**) perceived by the dairy farmers to ensure revenue. To our knowledge this question is not present in other studies. Moreover, respondent producers were asked not to take into account their current farm when considering their IFF. The data collection was conducted more recently, at the end of 2014 and the beginning of 2015. This was a particular context, just before the quota removal, when producers had this new perspective in mind. This change implied the disappearance of regulation of dairy supplies and was bringing uncertainty about the milk price [16]. We have assumed that this change in their working framework impacted respondents reflections and led them to reconsider their strategies, taking into account this new reality. They had just faced two important milk crises associated with low milk price in 2009 and an increase of the cost of inputs in 2012. This research studied unprecedented reasons for the choice of IFF compared to what is present in the literature, to our knowledge, such as past events of the farms. Moreover, the present study explored the environmental and training aspects linked to this IFF vision. The environmental aspect is of high importance at a time of increasing awareness of the impacts of agriculture and breeding on the environment such as carbon footprints, biodiversity, etc. The topic of trainings for dairy producers was studied to orientate universities and other stakeholders of breeding research and

development towards the domains needed and desired by dairy producers. A comparison between the current farm and the IFF of the respondent was realised, and permitted the difference between the reality and the aspiration of the producers to be studied. More specifically, the goals of this study were to answer the following questions: (1) What is the perception of dairy producers of their IFF? (2) How do dairy producers distribute themselves between IFF highlighted? By gathering different kinds of information, of which some are novel or rarely present in the literature, this study also answered to the following questions: (3) How do farmers decide on their IFF? (4) How do environmental aspects factor into IFF decisions? (5) Which paths and themes for training do farmers want in order to reach their desired IFF? And, ultimately, (6) how do farmers' IFF compare to their current dairy farming systems?

## Materials and methods

### Survey and IFF typology

In 2014, moving towards the end of the quota, as stakeholders of the dairy sector (research centre, agricultural sciences faculty, breeding association, agricultural unions, etc.), we wanted to know how the dairy producers of the southern part of Belgium will react to this change. We created a survey using LimeSurvey software (version 3.15.1+181017, LimeSurvey GmbH, Hamburg, Germany), which provides an internet link to get access and to complete the survey. The survey was first pre-tested orally with two dairy producers to estimate its duration and its clarity. The Board of Ethics and Scientific Integrity of the University of Liege waives the need for ethical approval. We communicated with Walloon dairy producers about the goals of the survey and its access broadly via all communication ways towards them: specialised press, agricultural internet websites, Unions and also advertisements through the milk payment letter which is sent to all the Walloon dairy producers once a month. The survey written in French can be viewed at the following internet link: https://www.gembloux.ulg.ac.be/enquete/index.php/219425?lang=fr

A total of 245 producers completed our survey between November 2014 and January 2015.

The entire survey was composed of 127 questions where the answers were decomposed into 498 categorical and 44 quantitative variables.

The question 'Without taking into account your current farm, what is, according to you, the ideal future farm to ensure a revenue?" was proposed to the producers and they must choose between short propositions on seven items: 1) intensive or extensive production; 2) specialised or. diversified activity (or activities); 3) farming strongly or weakly based on new technologies; 4) farm managed by an independent farmer or a group of managers; 5) family or employed workforce; 6) providing production for local or global markets; 7) providing standard or differentiated quality production. The modality "no opinion" was available for each IFF question. Counts were calculated for all modalities of these seven sub-questions.

The first step was to study if there were relationships between all modalities derived from the seven sub-questions asked. To achieve this objective, a multiple correspondence analysis (**MCA**) was carried out as the variables were categorical. For a MCA, the eigenvalue of the dimensions generated, named principal inertia, is a biased measure of the amount of information presented by a dimension [17]. Corrected inertia rates were calculated, as described by Benzécri [18], to quantify the correct proportion of information of a dimension.

Classes were established to study the distribution of producers along the dimensions of the MCA. The interval between the 1% percentile and the 99% percentile of each dimension was divided equally into five classes. Then, the individuals per class were counted.

To exclude a group of producers with some characteristics if necessary, cluster analysis with the WARD method was used on the scores of the individuals on each dimension of the MCA. The WARD method is a hierarchical agglomerative method [19]. The principle of this kind of

method is to put initially the n individuals in n groups and then to agglomerate the groups. The algorithm of WARD makes it in such a way that the gatherings induce the lowest decrease of $R^2$ at each step.

If a group of producers was excluded, its characteristics were previously studied against the remaining producers. The level of significance of the difference of the quantitative characteristics between the excluded and the remaining producers was studied thanks to general linear models. The level of significance of the difference of the proportions for each modalities of the qualitative characteristics between the excluded and the remaining producers was studied using tests of proportions.

### Characterisation of IFF choice

To describe the dairy producers in terms of their IFF, the scores on MCA dimensions were studied as a function of other variables extracted from the survey. This method was chosen instead of the creation of classes, possible with the Latent Class Analysis method or the Numerical Classification on the scores of MCA (Hierarchical Clustering on Principal Components). This choice was motivated by the wish to not put the producers in boxes but to study their position on a gradient between potential extreme models identified along the dimension.

The other variables extracted from the survey whose the relationships with the dimension are studied were distributed within several themes. These were the effect of past crises, problems encountered by the farmer, production factors, age of the farmer, breed of the cow, diversification of activities and alternative valorisation, regrouping between producers, consideration of mechanisation and robotisation on the farm, the reaction of the farmer to external factors, the considerations of farmers about environmental aspects, climatic hazards, ways to reach the ideal formation and field of formation. For categorical variables, the scores of MCA dimensions were modelled using these variables as a fixed effect in a generalised linear model. Least squares means were estimated for the two-by-two comparisons using the Tukey test. The level of significance of those differences was assessed based on the *P*-value of the test. For quantitative variables, Pearson correlation coefficients were calculated between the scores of MCA dimensions and these variables. Their corresponding *P*-values were estimated to observe if the correlation values were significantly different from 0.

To observe if dairy producers presented the farming characteristics they considered to be ideal at the moment of survey, absolute frequencies (counts) were calculated as a function of each ideal future farm characteristic and of the answer to the question which corresponds to this characteristic for the current situation (Table 1). Moreover, the percentage of "unhappy" producers was calculated as the ratio between the producers not currently in the situation that they consider as ideal and the total number of producers.

All editing and statistical analyses were carried out using SAS software (version 9.4., SAS Inst. Inc., Cary, NC, USA).

## Results and discussion

### Data representativeness

A total of 245 producers answered the survey, giving a response rate of 6.1% (about 4,000 dairy producers in 2015 and 3,500 in 2017 in Wallonia [4]). The density of dairy farms throughout Wallonia was well represented in the sample, with a higher answer rate in the provinces more populated with dairy farms. More answers were obtained in the east part of Wallonia, where a higher density of dairy farms exists due to the grazing landscape that is particularly suitable for dairy production. Wallonia is a highly heterogeneous region with regard to soil and geological characteristics [20].

**Table 1. Absolute frequencies (counts) of producers as a function of their answer to the ideal future farm characteristic and the corresponding characteristic for the current situation and percentage of "unhappy" producers (*i.e.*, percentage of producers not currently in the situation that they consider to be ideal) (N = 245).**

| | | Corresponding characteristic for the current situation | | % of "unhappy" producers |
|---|---|---|---|---|
| | | >2 cows per hectare of grass | <2 cows per hectare of grass | |
| **Ideal future farm characteristic** | Intensive | 38 | 66[1] | 50% |
| | Extensive | 22 | 51 | |
| | | Only dairy production activity | Presence of activities other than dairy production | |
| | Specialised | 46 | 59 | 37% |
| | Diversified | 23 | 93 | |
| | | Presence of milking robot or agricultural equipment for a better technicality | Absence of milking robot or agricultural equipment for a better technicality | |
| | Strongly based on new technologies | 33 | 52 | 37% |
| | Weakly based on new technologies | 16 | 85 | |
| | | >1 chief operating officer or associates | 1 chief operating officer | |
| | Managed by a group of managers | 20 | 25 | 42% |
| | Managed by an independent farmer | 68 | 108 | |
| | | Presence of workers (i.e., external person to family working on the farm) | No workers | |
| | With family workforce | 17 | 195 | 10% |
| | With employed workforce | 7 | 6 | |
| | Providing dairy production for local *vs.* global market | No corresponding characteristic | | |
| | Providing standard *vs.* differentiated quality dairy production | No corresponding characteristic | | |

[1] Frequency in grey box corresponds to producers not currently in the situation that they consider as ideal regarding this characteristic

Dairy producers of the survey declared a mean of 79 cows and 86ha. Dairy production was their unique activity for 33% of them. The mean number of dairy cows per Walloon dairy farm was 52.9 cows in 2015 [21]. No regional statistic exists on the mean agricultural area of all producers that perform a dairy activity. The mean agricultural area of specialised dairy farms and of all kind of farms taken together equated to 61.98 ha and 55.8 ha, respectively [21]. So, the producers surveyed tended to have bigger farms regarding herd size and agricultural area than the average Walloon farm that have dairy activity.

## What is the perception of dairy producers of their ideal future farm?

**Univariate approach.** As mentioned previously, the first aim of this study was to highlight the perceptions of Walloon dairy producers of their ideal farm, just before the end of the milk quota. This was done through the answers to 7 sub-questions. Table 2 shows the frequency for each modality of those questions.

Contrasting opinions of dairy farmers were observed for almost all questions except for the type of management and the kind of workforce: 71.84% of the respondents wanted an independent farmer management, and 86.53% focused on a family workforce (Table 2). These results highlight a will in the southern part of Belgium to maintain the traditional structure of work organisation in the future, with family workforce and one director of operations. More globally in the world, dairy farms are still mostly owned and managed by a family structure,

**Table 2.  Percentages of responses to the seven questions about the ideal future farm (N = 245).**

| Question | Proposition | Percentage (%) |
|---|---|---|
| **Without taking into account your current farm, what is, according to you, the ideal future farm to ensure a revenue?"** | | |
| Intensive *vs.* extensive | Intensive | 43 |
| | Extensive | 30 |
| | No opinion | 27 |
| Specialised *vs.* diversified | Specialised | 43 |
| | Diversified | 47 |
| | No opinion | 10 |
| Strongly *vs.* weakly based on new technologies | Strongly | 35 |
| | Weakly | 41 |
| | No opinion | 24 |
| Managed by an independent farmer *vs.* a group of managers | Independent farmer | 72 |
| | Group of managers | 18 |
| | No opinion | 10 |
| Family *vs.* employed workforce | Family | 87 |
| | Employed | 5 |
| | No opinion | 8 |
| Providing dairy production for local *vs.* global market | Global | 43 |
| | Local | 32 |
| | No opinion | 25 |
| Providing standard *vs.* differentiated quality dairy production | Standard | 38 |
| | Differentiated quality | 45 |
| | No opinion | 17 |

whatever the degree of development of the country [22, 23]. The choice of producers to work by themselves and not to deal with workers (i.e., an external person to the family employed on the farm) was noted in other studies. For example in Spain Gonzalez and Gomez [24] observed, when asking 3,370 farmers for their definition of a farmer, that more than half of them chose labourer and 12% chose businessman. In the USA in 1988, Mooney presented the fact that farmers had a particular status, being workers and employing other workers [25].

From Table 2, it is interesting to note that the highest percentages of abstention were observed for the questions about intensive *vs.* extensive, strongly *vs.* weakly based on new technologies, and providing DP for local *vs.* global markets. These results show that a quite significant proportion of the respondents did not take a position on these directions for the evolution of dairy farms.

**Multivariate approach.**  To study the relationships between the answers given by the respondents to all questions about IFF, a MCA was performed as the related variables were categorical (Table 2). The percentage of principal inertia of the dimensions 1 and 2 of MCA were 16.75% and 12.38%, respectively (Fig 1). The value of corrected inertia for the two first dimensions reached 72.7% and 21.5% respectively, gathering almost 95% of the information.

The modalities no opinion of each characteristic showed positive scores on the first dimension of MCA and the modalities with an opinion showed negative scores. Thus, the first dimension of the MCA allowed differentiation between the producers who did not give their opinion concerning characteristics of IFF and the producers who did (Fig 1). Cluster analysis was used to isolate the group of producers with a lot of 'no opinion' answers to the seven

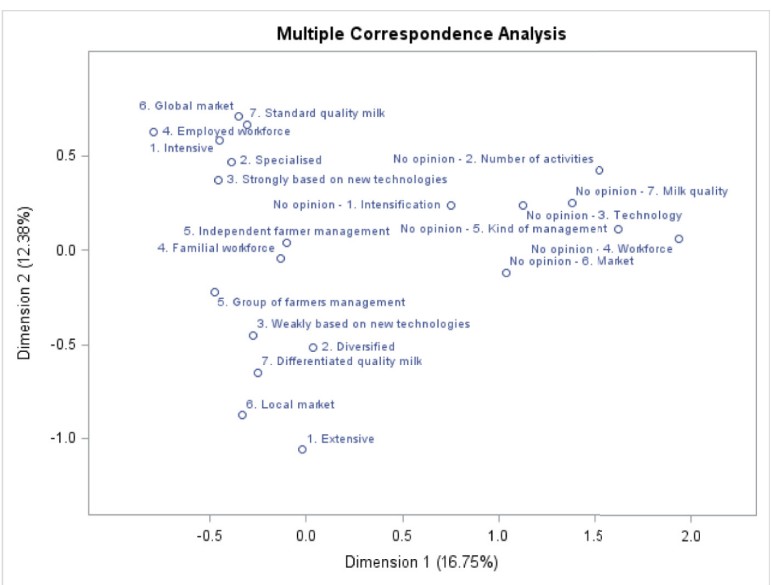

**Fig 1. Representation of the modalities in the multiple correspondence analysis first factorial plan.** Values of principal inertia reached 16.75% and 12.38%. Values of corrected inertia reached 72.7% and 21.5%.

questions: this formed the first separation of classes of the analysis, dividing the "no-opinion" producers (15%) from the others (85%). The no opinion producers cluster (N = 38) was removed from the analysis to avoid potential bias coming from farmers who did not have a clear vision of their IFF.

Detailed information about this group is available in Tables 3 and 4. They tended to be older farmers (45–54 years), who came from Liège, which is a historic dairy region (Table 3). Percentages of grass and corn silage observed for this group highlighted a same way of feeding as the complete sample (Table 4). Even if these differences are not significant ($P = 0.20$, $P = 0.59$, $P = 0.33$), the more represented single breed and the lower number of cows but with the higher milk delivery quota of the no-opinion producers tended to express quite technical and high performing producers in this group. They seemed to be people who have been dairy producers for a long time. We could assume that their farms had good economic performances and did not lead them to think about evolution in response to a great change (*i.e.* the quota removal).

The second dimension of the MCA showed positive relationships with some modalities of the IFF characteristic and negative relationships with their opposite. Thus, this dimension seems to highlight the wishes of dairy farmers about their IFF, for those who took a position on this question. More precisely, this axis showed a gradation of question modalities and proximity between several characteristics. The second dimension of the MCA was the most interesting for highlighting the wishes of dairy farmers about their IFF, for those who took a position on this question. This axis showed a gradation of question modalities and proximity between several characteristics. This dimension led to the identification of two extreme tendencies (Fig 1); the modalities of familial workforce, independent farmer management and management by a group of farmers were near to zero on this axis (Fig 1). This means that the small proportion of producers supporting group management was distributed between the two extreme tendencies observed. The position of the modalities of familial workforce and independent farmer at the middle of the second dimension illustrated the fact that these modalities were chosen by producers from the two tendencies identified. The small proportion

**Table 3. Percentages of producers as a function of modalities of categorical variables for the no-opinion producers and the complete sample.**

| % | Complete sample N = 245 | No-opinion producers N = 38 | Producers with an opinion N = 207 | P No-opinion *vs.* with opinion |
|---|---|---|---|---|
| **Age (years)** | | | | |
| 0–34 | 18 | 16 | 19 | 0.64 |
| 35–44 | 31 | 32 | 31 | 0.98 |
| 45–54 | 37 | 47 | 35 | 0.15 |
| 55–64 | 13 | 6 | 15 | 0.027 |
| **Geographical situation** | | | | |
| Walloon Brabant | 6 | 2 | 6 | 0.24 |
| Hainaut | 32 | 34 | 31 | 0.74 |
| Liège | 34 | 45 | 32 | 0.14 |
| Luxembourg | 10 | 5 | 11 | 0.20 |
| Namur | 19 | 13 | 20 | 0.28 |
| **Importance of dairy activity** | | | | |
| Unique activity | 33 | 34 | 32 | 0.83 |
| Preponderant activity | 65 | 61 | 65 | 0.59 |
| Secondary activity | 3 | 5 | 2 | 0.45 |
| **Herd breed** | | | | |
| Single breed | 33 | 42 | 31 | 0.20 |
| Multi-breed | 67 | 58 | 69 | 0.20 |
| **Other animal production** | | | | |
| Yes | 45 | 37 | 46 | 0.27 |
| No | 55 | 63 | 54 | 0.27 |
| **Milk production evolution in the next 5 years** | | | | |
| Decrease | 2 | 0 | 2 | 0.043 |
| Constant | 54 | 63 | 52 | 0.20 |
| Increase | 38 | 32 | 40 | 0.33 |
| Stop | 6 | 5 | 6 | 0.79 |
| **Agricultural area investment since 2009** | | | | |
| Yes | 46 | 47 | 46 | 0.88 |
| No | 54 | 53 | 54 | 0.88 |
| **Agricultural area investment in the next five years** | | | | |
| Yes | 57 | 64 | 55 | 0.27 |
| No | 43 | 36 | 45 | 0.27 |

of producers choosing an employed workforce was positioned at the top of the second dimension (Fig 1).

The first tendency, related to high scores on the second MCA dimension, corresponds to IFF with the following characteristics: global market, standard milk, intensive system, employed workforce, specialised and strongly based on new technologies. Other authors have

**Table 4. Means of quantitative variables for no-opinion producers and the complete sample.**

| | Complete sample N = 245 | No-opinion producers N = 38 | Producers with an opinion N = 207 | P No-opinion *vs.* with opinion |
|---|---|---|---|---|
| Agricultural area (ha) | 86 | 87 | 85 | 0.80 |
| Percentage of corn silage | 15 | 14 | 15 | 0.67 |
| Percentage of grass | 61 | 60 | 60 | 0.76 |
| Milk delivery quota (l) | 558743 | 632880 | 545133 | 0.33 |
| Number of cows | 79 | 73 | 81 | 0.59 |

observed the same relations. From a trial of 458 French dairy farms, Hostiou *et al.* [26] highlighted a profile of farmers which simultaneously gathered high equipment, intensification and workers. From a trial of 3,370 producers of all sectors in Spain, Gonzalez and Gomez Benito [24] collated the characteristics of large holdings, market-orientated farming and management of workers. Cournut *et al.* [27] highlighted different ways of evolving dairy farming in France, characterised by workers, mechanisation and high equipment. This tendency in dairy farming systems is explained by the evolution of the dairy system [6]. The increased competition in the dairy market caused by the creation of the open European market, as well as the wish of consumers to have structures that gather all the food supplies in one place (*i.e.* a supermarket) led to the concentration of dairy processing in a few big firms [11]. These firms were better placed to develop because they could control their collection costs, benefit from scale economies and were able to deliver to supermarkets with regularity in quantity and with a standard quality [9]. This state and the world market have conditioned milk prices for the producers. Increasing production, thanks to more cows or higher productivity, is a possible way to stay profitable, considering the undergone milk price [5, 11]. To achieve profitability, an elevated production of milk per cow and an increase of cows on the farm are reached [11]. Moreover, this increase in milk production at farm level was also forced by the orientated production Common Agricultural Policy (**CAP**) primes, although CAP has limited help for the dairy sector. Therefore, all of these characteristics intensify the dairy farming system. Intensification was defined by Garcia-Martinez *et al.* [28] as the maximisation of the rarest factor, traditionally the agricultural area. The increase in DP per unit of agricultural area was possible thanks to intensive production of forage and purchase of inputs that are produced where production costs were the lowest, to balance the ration and to increase the production per cow, or the number of cows reared on a hectare of agricultural area and therefore DP per unit of agricultural area at the level of the farm [9, 11]. This intensification led to more specialised farms with more dairy cows and their entire workforce directed to this specialisation [9]. The enlargement of farms required a higher work rate; this was surmounted thanks to equipment and new technologies and to increased human workforce: collective organisation, subcontracting to private firms and employment of workers [9].

The second tendency, contrary to the first tendency, was characterised by high negative scores on the second MCA dimension. This axis was represented by the following modalities: weakly based on new technologies, diversified, differentiated quality milk, local market and extensive system (Fig 1). This reflects another form of dairy farming. This form is favoured by a constant increase in input prices, combined with a growing demand from consumers for high quality and local-based products [9]. These dairy producers choose to work with greater self-sufficiency to be less dependent on the undergone input prices [9]. The "localisation" of the production demanded by consumers was executed thanks to this more locally-produced forage and fewer inputs from outside [5]. This return to self-sufficiency led to more extensive farming [5]. The production induced was also often quality-differentiated and dedicated to local markets [9]. Cournut *et al.* [9] showed in their study that this kind of dairy farming is chosen by a minority of farms, which are still diversified.

This gradation with two kinds of models at the extremities of the second MCA dimension was also described in other studies [5, 6, 9, 11, 29–31]. They were named globalisation *vs.* territorialisation by Cournut *et al.* [9], or globalisation *vs.* localisation by Napoleone *et al.* [11]. Lebacq [6] identified a "dualisation of dairy farming systems between 'a mainstream model' focusing on an increasing farm size, production intensity and specialisation and alternative models involving initiatives deviating from this trend and constituting niche developments (niches = minor elements, hardly sustainable against the mainstream model)". Thanks to a survey answered by 180 producers of all sectors in 2007 in France, concerning the evolution of

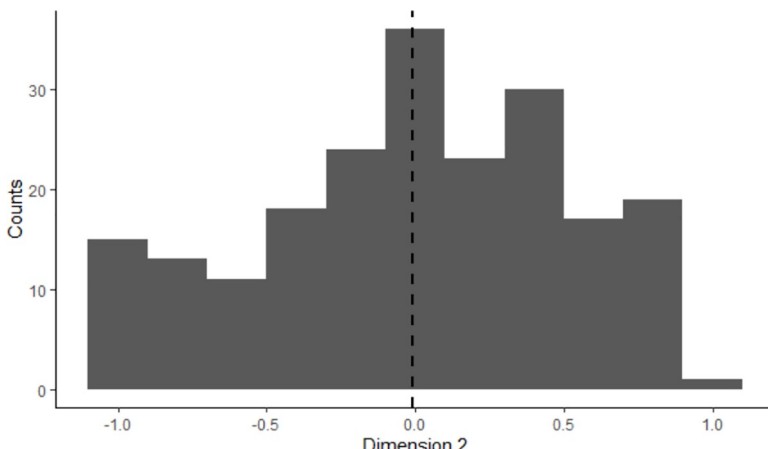

**Fig 2. Distribution of the producers along the second dimension (the dotted line represents the mean score on the second dimension of the producers) (N = 207).**

their farms and their aspirations, Dockès *et al.* [30] also highlighted a major tendency towards the enlargement, professionalisation and specialisation of farms, but those authors also mentioned that other farms wanted to develop diversified structures, orientated towards the requests of society, processing and farm accommodation.

## How do dairy producers distribute themselves between the ideal future farms highlighted?

The present study showed a bifurcation and quantified two ways: 46% *vs.* 26% of producers having high positive and high negative scores respectively on the second dimension (Fig 2). Verhees *et al.* [15] quantified producers as a function of their strategies of development, but solely regarding specialisation *vs.* diversification of their activity, 54.3% *vs* 15.1% respectively. The bifurcation phenomenon is also observed in the organic sector. Two models appeared: organic agriculture realised by historic actors and the other driven by the agribusiness to answer to a increasing organic demand [32–34].

## How do farmers decide on their ideal future farm?

To study the relationships between the different IFF, the reasons for these and other interesting technico-economic information, the second dimension was considered as a gradient (**IFFg**) interpreted at the extremities as global-based intensive producers (GBI: high positive scores) and local-based extensive producers (LBE: high negative scores). The choice to work with a gradient rather than a clear separation of the two tendencies was motivated by the will to not put dairy producers into boxes. The mean of the scores of the second MCA dimension was –0.012 with a SD of 0.053. Minimal and maximal values were –1.09 and 0.92, respectively.

Based on the interpretation of IFFg, a significant negative correlation indicates a higher relationship with the dairy producers desiring a LBE model. By opposition, a significant positive correlation means a higher link with the dairy producers desiring a GBI model. Tables 5, 7 and 8 give the results of generalised linear models where the categorical variables were introduced separately as a fixed effect in the model. Significantly lower estimates of IFFg for a specific modality of the considered categorical variable depicts a tendency of producers desiring a LBE model to choose this modality, while significantly higher estimates of IFFg means a tendency of producers wanting a GBI model to choose this modality. These analyses were

**Table 5. Value and level of significance of the difference in the ideal future farm gradient as a function of modalities of categorical variables: Reasons (N = 207).**

| Categorical variable | Modality and estimate | | | | P |
|---|---|---|---|---|---|
| **Past crisis** | | | | | |
| Presence of deep modifications after crisis | No | | Yes | | 0.025 |
| | 0.031 | | −0.17 | | |
| **Workload** | | | | | |
| Degree of arduousness | Not arduous | Arduous | | Highly arduous | 0.0043 |
| | −0.11[b] | −0.092[b] | | 0.15[a] | |
| Member of an agricultural replacement service | Yes | | No | | 0.059 |
| | 0.058 | | −0.0801 | | |
| Worker engagement: help for workload and administrative aspects | Already implemented 0.024[ab] | To implement in the future 0.13[a] | | Not interested −0.040[b] | 0.25 |
| **Production factors** | | | | | |
| Milk production evolution for 5 years | Decrease | Constant | | Increase | 0.036 |
| | −0.21 | −0.13 | | 0.052 | |
| Agricultural area investment since 2009 | No | | Yes | | 0.0002 |
| | −0.14 | | 0.14 | | |
| Investment (no agricultural area) since 2009 | No | | Yes | | 0.055 |
| | −0.19 | | 0.015 | | |
| Agricultural area investment in 5 years | No | | Yes | | 0.073 |
| | −0.0909 | | 0.046 | | |
| Geographical situation | Brabant Walloon0.11 | Liège 0.12 | Namur 0.055 | Luxem- bourg −0.013 · Hainaut −0.099 | 0.51 |
| **Age** | | | | | |
| Age (years) | 0–34−0.086 | 35–44 0.0105 | 45–54 0.0049 | 55–64 0.0502 | 0.67 |
| **Diversification and alternative valorisation** | | | | | |
| Presence of other animal production | Yes | | No | | 0.037 |
| | −0.093 | | 0.0603 | | |
| Dairy processing and direct sales: sector developed if supported | Yes | | No | | <0.001 |
| | −0.33 | | 0.11 | | |
| Processing and direct sales (except dairy): sector developed if supported | Yes | | No | | 0.0096 |
| | −0.39 | | 0.013 | | |
| HORECA, tourism and teaching activity to develop even if sustained | Yes | | No | | 0.055 |
| | −0.18 | | 0.018 | | |
| Concern for diversification | Yes | | No | | <0.001 |
| | −0.23 | | 0.17 | | |
| Alternative chain for milk production valorisation | Yes | | No | | <0.001 |
| | −0.49 | | 0.036 | | |
| Alternative chain for "other than dairy" activity | Yes | | No | | 0.0017 |
| | −0.56 | | −0.0087 | | |
| Increase of "other than dairy" activity without investment | Yes | | No | | 0.012 |
| | −0.42 | | −0.013 | | |
| No activity to develop even if sustained | Yes | | No | | <0.001 |
| | 0.27 | | −0.10 | | |
| Increase of added value in farms: advantage of diversification and transformation | Yes | | No | | 0.0047 |
| | −0.097 | | 0.11 | | |
| Link between producers and consumers: advantage of diversification and transformation | Yes | | No | | 0.0064 |
| | −0.14 | | 0.066 | | |

*(Continued)*

**Table 5.** (Continued)

| Categorical variable | Modality and estimate | | | P |
|---|---|---|---|---|
| Conservation of farms in the region: advantage of diversification and transformation | Yes | | No | 0.037 |
| | −0.16 | | 0.028 | |
| Financial, decisional and technical autonomy: advantage of diversification and transformation | Yes | | No | 0.005 |
| | −0.27 | | 0.030 | |
| Consumer loyalty: limit to diversification and transformation | Yes | | No | 0.028 |
| | 0.17 | | −0.047 | |
| Regulatory constraints (hygiene, *etc.*): limit to diversification and transformation | Yes | | No | 0.065 |
| | −0.080 | | 0.055 | |
| Size of investments: limit to transformation and diversification | Yes | | No | 0.030 |
| | −0.14 | | 0.0401 | |
| No constraints for transformation and diversification | Yes | | No | 0.052 |
| | −0.093 | | 0.042 | |
| No advantage of diversification and transformation | Yes | | No | 0.0006 |
| | 0.27 | | −0.066 | |
| **Breed** | | | | |
| Composition of the herd | Single breed | | Multi-breed | 0.0005 |
| | 0.18 | | −0.095 | |
| | Pure-bred | | Dual purpose breed: | 0.0023 |
| | 0.0058 | | −0.19 | |
| **Regrouping** | | | | |
| Advantage of fiscal and administrative aspects: advantage of grouping | Yes | | No | 0.050 |
| | −0.16 | | 0.023 | |
| Development of a joint project: advantage of grouping | Yes | | No | 0.072 |
| | −0.15 | | 0.020 | |
| Better marketing of the products: advantage of grouping | Yes | | No | 0.030 |
| | −0.38 | | 0.0063 | |
| **Mechanisation and robotization** | | | | |
| Mechanisation, robotisation: help with workload and administrative aspects | Into effect−0.031[ab] | Not interested −0.094[b] | To activate 0.12[a] | 0.041 |
| **Reaction to external factors** | | | | |
| Arduousness of the economic uncertainty of input price | No | | Yes | 0.0089 |
| | −0.19 | | 0.0403 | |
| Will of a dairy factory imposing production limits | Yes | No | Not important 0.0088[b] | <0.001 |
| | −0.23[c] | 0.25[a] | | |
| Evolution of milk production during crises (2009, 2012) | Decrease | No variation | Increase | 0.0006 |
| | −0.17[b] | −0.092[b] | 0.21[a] | |
| Ideal size of the dairy factory | Small | Medium | Large 0.42[a]   Not important −0.026[b] | <0.001 |
| | −0.52[c] | −0.11[b] | | |

Means with different letters are significantly different.

conducted on the producers who have an opinion (N = 207). The following paragraphs will summarise the potential reasons driving the choice of IFF made by the Walloon dairy farmers.

**Effect of past crisis on perceptions of the ideal future farm.** The producers that were impacted by past crises wished more for a LBE model (estimate = –0.17, Table 5). This could be related to the suffering involved in the crisis and the wish to apply solutions in order to not repeat this situation: revenue from diversified activities, other outlets for the milk production sold (*i.e.* local market characteristic) and/or self-sufficiency to be less dependent on purchased feed (*i.e.* extensive farm characteristic). This is in agreement with a past finding [35]. We observed a decrease in intensification in 2012 which was the year of a dairy economic crisis mainly related to an increase in the price of inputs.

**Workload.** Workload seems to be less bearable for producers desiring a GBI model (estimate = 0.15, Table 5; $R_{\text{workforce constraint}}$ = 0.22, $P$ = 0.002). Producers wishing for a GBI model were also more likely to be members of an agricultural replacement service (estimate = 0.058, Table 5) and showed a tendency to be more interested in employment of workers (estimate = 0.13, $P_{\text{worker engagement to implement } vs. \text{ not interested}}$ = 0.11, Table 5). The choice of GBI model could be explained by this current workload, involving the need for an increase of revenue. So, the solution considered could be higher milk production and the breeding of more cows rather than diversification of activities and self-valorisation activity, the development of which requires a lot of time. Samson *et al.* [36] confirmed this in the Netherlands by highlighting a nearly significant effect of labour productivity on the DP increase strategy.

**Production factors.** The size of agricultural area, the milk delivery quota, the number of cows and the percentage of corn silage currently observed in the farming system showed significant and positive correlations with IFFg (R = 0.15, 0.36, 0.18 and 0.24; $P$ = 0.033, <0.001, 0.0099, 0.0002 respectively). So, dairy producers choose their IFF partly as a function of their current production factors. This is expected as a higher number of hectares, cows and litres means a higher capacity of the dairy installation, of the material and so the possibility of a more preponderant dairy activity. The higher percentage of corn silage also reflected the possibility to seed corn silage, allowing the intensification of production as required within a GBI model. Similar relationships between characteristics of the farm and current or desired models of farming were observed by others. For Central and Eastern Europe, Verhees *et al.* [15] showed that land was the most important factor in developing a specific farming strategy. In France, Hostiou *et al.* [26] observed that intensified farms with higher technology equipment sometimes employed more workers, and were the farms with significantly higher agricultural area, percentage of corn silage, number of cows and milk quota. In the Netherlands, Samson *et al.* [36] showed that production intensity, number of cows, modernity of technology and availability of land were important factors in DP increase strategies.

In contrast, producers with lower production factors can consider rarely more enlargement and therefore think differently about the enhancement of their revenue: better valorisation of quality differentiated milk, other activities on the farm, self valorisation, the LBE model. Samson *et al.* [36] showed that lower stable capacity varies inversely to a DP increase strategy, which is rather a GBI tendency.

The findings of the current study, as confirmed by previous researchers, showed that producers work within a tightly constrained and regulated environment limiting their ability to determine the future of their farm according to their personal desires. This statement was also concluded by Mc Elwee *et al.* [37] and Methorst *et al.* [38]. In the Netherlands, Keizer and Emvalomatis [39] and Groeneveld *et al.* [40] showed that bigger farms are more likely to increase than other farms.

However, based on the quite low values of the correlations obtained between the agricultural area and the number of cows, we can consider that this situation must be nuanced and that the IFF chosen also depends on the opinions of the dairy producer, not taking into account the current situation of his farm. This statement is reinforced by the fact that the

correlation of percentage of meadow with IFFg was not significantly different to 0 (R = −0.097, $P > 0.1$). Also, the impact of the provinces of the Walloon Region, which present different geographical and soil characteristics, on IFFg were not significantly different ($P = 0.51$, Table 5).

Moreover the significant relations between IFFg and milk production evolution for five years (Table 5; $R_{\text{quantity of milk variation}} = 0.30$, $P < 0.001$), investment for and in five years (Table 5) support the assumption that the IFF chosen depends greatly on the mentality of the producers.

In their study, Methorst *et al.* [13] proved the heterogeneity of farm developments of producers facing the same socio-material context, showing the importance of the mentality of the producers in their decisions. Authors speak about shared values, norms, ways they see themselves or would like to be seen by producers, views, capacities and their perceptions of opportunities and any room for manoeuvre, skills, motives, entrepreneurship, goals and strategies [12, 13, 36, 38, 41] as factors which influence farm development. Samson *et al.* [36] discussed experimental economics, which are economics where psychology and biology, which explain human behaviours, are added to better explain the development of enterprises. The consideration of more than just economic aspects permits them to reduce the error of their model for predicting DP increase strategies [36].

**Age.**    Age of the producer seems not to condition the desired IFF (Table 5). An IFF could be chosen because of either the new ideas of young producers or the experience of older producers. If mentality seems to influence IFF choice, it is not linked to age. The two kinds of IFF could be an answer to both innovation and problems encountered during a long career. Samson *et al.* [36] also studied age as a reflection of the farmers' values, goals and strategies, and showed no relationship with DP increase, which is rather a GBI characteristic. On the contrary, on the basis of data from 11 countries of the European Union, Weltin *et al.* [42] observed an effect of age on the tendency towards diversification, which is rather a LBE tendency.

**Diversification and alternative valorization.**    The results obtained in this study showed a link between the diversification mentality and the choice of LBE model. Significant negative estimates or correlations were observed for the following variables related to diversification: the presence of other animal production (estimate = −0.093, Table 5); the direct selling milk quota (R = −0.17, $P = 0.016$); dairy or no dairy processing and direct sales (estimates = −0.33 and −0.39, Table 5); the development of HORECA activities, tourism and teaching (estimate = −0.18, Table 5); the concern for diversification (estimate = −0.23, Table 5); alternative chain for milk and other than milk production valorisation (estimates = −0.49 and −0.56, Table 5) and the increase of "other than dairy" activity without investment (estimate = −0.42, Table 5). Conversely, producers desiring a GBI model were more likely to choose the item "no activity to develop if supported", suggesting the unique principal activity way of thinking of producers aiming for a GBI model (estimate = 0.27, Table 5). Samson *et al.* [36] confirmed this tendency and showed that the presence of diversified activities evolved inversely to the increase of milk production. In this study, we observed potential explanations to support to this fact. Producers wishing for a LBE model considered self-valorisation and diversification as solutions to the current situation to enhance revenue due to the creation of added value (estimate = −0.097, Table 5). They thought that diversification and transformation allowed financial, decisional and technical autonomy (estimate = −0.27, Table 5) and were confident in consumer loyalty (estimate = −0.047, Table 5). They considered relations with consumers as an opportunity and not a threat, unlike producers desiring a GBI model (estimate = 0.17, Table 5). One reason GBI model producers gave against self-valorisation and diversification seemed to be the lack of trust in consumers and therefore the outlets. They frequently saw no advantage to self-valorisation and diversification (estimate = 0.27, Table 5). The relation to the consumer was also

studied by Verhees *et al*. [15]. They observed that consumer orientation was more often declared as an opportunity by the profiles of producers considering strategies similar to LBE. The positive impact of diversified activities on autonomy was also shown by Bergevoet *et al*. [12]. They mentioned that proponents of the "extra source of income" model (closest to the LBE model) were more able to declare that they can increase the sales-price of their milk. Producers wishing for a LBE model were also likely to find no constraints to transformation and diversification (estimate = −0.093, Table 5). The only limits to diversification and transformation highlighted by producers wanting a LBE model were regulatory constraints (estimate = −0.080, Table 5) and the size of investments (estimate = −0.14, Table 5). As a consequence of these considerations, producers wanting a LBE model felt that they were more able to meet society's expectations regarding local and artisanal products (R = −0.22, *P* = 0.0016) and the desire for a familial structure (R = −0.12; *P* = 0.084).

**Breed to produce milk.** Producers wanting a LBE model are more open to breeding a dual-purpose herd (estimate = −0.19, Table 5), which permits them to diversify their production: milk and meat. Producers wishing for a GBI model target a single, more specialised breed (estimate = 0.18, Table 5) which could offer more homogeneous management of the herd. The link between mentality, observed through the choice of breed(s), and the choice of IFF is once more highlighted.

**Regrouping.** Producers tending towards the LBE model were more likely to promote regrouping for its advantages regarding fiscal and administrative aspects, the development of a joint project and the marketing of the products (estimates = −0.16; −0.15; −0.38, Table 5). The importance of mentality for the choice of IFF has been shown. A mentality of cooperation, as a solution to enhance their quality of life and revenue, tends to be shared between producers desiring a LBE model.

**Mechanisation and robotisation.** We observed that the wish of technology of producers tending towards GBI model can be explained by the fact that they considered it as help for workload (estimate = 0.12, Table 5). It can be assumed that the solution considered by them is to keep the same activity or increase it with help from machines. In southern France, Dufour *et al*. [43] observed the propensity of farmers with workers, close to the GBI model, to prioritise investment in equipment. Verhees *et al*. [15] observed that better management, including new technologies, was more cited as an objective for producers whose strategy profiles were more similar to the GBI than LBE models.

**Reaction to external factors.** Reactions of dairy producers to factors external to their decision-making power tend to be different as a function of their choice of IFF, showing once more a different mentality of the producers. Producers wanting a LBE model tend to show themselves to be more independent from the external economic actors: from the input producing companies (estimate = −0.19, Table 5) and from the market and factories, rejecting contracts which would link them to it (R = −0.13, Table 6). When their opinion about dairy factories was surveyed, producers desiring a LBE model preferred small or medium units with production limits (estimates = −0.52; −0.11; −0.23, Table 5), as before, which means regulation of the dairy offerings on the market. Producers wishing for a GBI model direct themselves to big units of processing without production limits (estimates = 0.42; 0.25, Table 5) and so more turned towards world markets. They recognise the freedom in regarding DP as an asset of quota removal (R = 0.23, Table 6). The reaction regarding the quantity of production was not similar during a crisis, producers wanting a LBE model tended to maintain or decrease their production (estimates = −0.17; −0.092, Table 5), whereas producers desiring a GBI model tended to increase production (estimate = 0.21, Table 5). The latter wanted to keep revenues constant with more litres produced when the price decreased, while the others controlled or decreased production when the gross margin per litre decreased. This can be due to a

**Table 6. Correlations (R) between the ideal future farm gradient and quantitative variables (N = 207).**

| Quantitative variable | R | P |
|---|---|---|
| **Reaction to external factors** | | |
| Contract means dairy production more integrated to dairy factories: level of agreement | −0.13 | 0.076 |
| Quota removal means more flexibility concerning production: level of agreement | 0.23 | 0.0014 |
| **Considerations of the environmental aspects** | | |
| Degree of the constraint: livestock manure application | 0.16 | 0.022 |
| Facility to answer to society's expectations: environmentally friendly agricultural practices | −0.15 | 0.027 |
| Agricultural activity is important for rurality of villages: level of agreement | −0.23 | 0.0011 |
| Agricultural activity is important for conservation of permanent grasslands: level of agreement | −0.27 | <0.001 |
| Agricultural activity is important for biodiversity: level of agreement | −0.18 | 0.0101 |
| Agricultural activity is important for planting and maintenance of hedges: level of agreement | −0.28 | <0.001 |
| Importance of answering society's expectations for the revenue of the dairy producers: level of agreement | −0.11 | 0.11 |
| Ease of answering society's expectations: landscape and territory maintenance: level of agreement | −0.19 | 0.0065 |
| **Needs: formation method** | | |
| *Frequency of calling replacement services for meeting and formations (N = 104) | 0.21 | 0.066 |

*producers declaring no calling of replacement services were removed from this analysis.

deliberate choice to decrease milk production or a decision to decrease the variable costs causing a decrease in milk production. These results can express a fear of producers tending toward the LBE model in considering world markets, contrary to producers tending towards the GBI model who have decided to work with this kind of market. Verhees *et al.* [15] observed that producers projecting strategies similar to the LBE model consider the market more as a threat than producers projecting strategies similar to the GBI model. Hansson *et al.* [44] and Weltin *et al.* [14] explained that this uncertainty and risk perception can explain the choice of diversification, which is a part of the strategy of the LBE model.

Couzy and Dockès [7] demonstrated different profiles of farmers and observed the entrepreneurship mentality of each one, which highlights similar tendencies to those presented here. Several profiles showed strong entrepreneurship but which was expressed differently to here. A category of farmers showed entrepreneurship by their wish for autonomy of decision in their management; they will keep a working approach close to the conventional one but with a modernist vision, always adapting to the market. They want to keep freedom in the classical framework. In 1988, Mooney described the split personality of producers: they are independent people, making their own decisions regarding their way of working and their investments but at the same time are people dependent on different processing actors and banks [45]. Another category of farmers showed entrepreneurship by their wish to develop an original idea, away from preexisting systems, a project in line with their conviction to be freer from the existing system [5].

Samson *et al.* [36] and Methorst *et al.* [13] reported that decisions of producers cannot be reduced to only economic aspects: this includes policies and market conditions but also their way of thinking about them.

## How do environmental aspects factor into IFF decisions?

The environmental aspects related to the desired IFF were studied as awareness of the environmental impact of breeding has become an important issue of our time.

Producers tending toward the GBI model seemed to work with a higher livestock manure application pressure (R = 0.16, Table 6) and therefore are already more likely to work in an intensified dairy system, which can have a greater impact on the environment. Samson *et al.* [36] showed a tendency toward manure production surplus by producers with increasing DP, which is rather a GBI characteristic.

Results of practices that are in accordance with the environment: measurement of the grass height, forage mixture with leguminous plants, use of a field notebook (estimates = −0.27; −0.11; −0.074, Table 7) showed a stronger interest from producers wanting a LBE model.

Besides these, all the significant negative correlations between IFFg and the levels of agreement with an agricultural area are important for the rurality of villages (R = −0.23, Table 6), for conservation of permanent grasslands (R = −0.27, Table 6), for biodiversity (R = −0.18, Table 6) and for hedges (R = −0.28, Table 6) showed the importance of the environment in the dairy activity of producers wanting a LBE model. It can be assumed that both LBE producers and GBI producers have concerns for the environment but in different ways. These results showed that LBE producers are more willing to employ the benefits of ecosystem services, which is observable in this database. Moreover, they found it easy to realise environmentally friendly agricultural practices, as asked for by society (R = −0.15, Table 6) and which are important to answer to society's expectations to guarantee their revenue (R = −0.11, Table 6).

Bergevoet *et al.* [12] had a considerably more consistent opinion. The "extra-source of income" profile producers (showing similarities with the LBE model) were more likely to declare that in their decision-making they take the environment into consideration, even if it lowers profit. The "large and modern farm" profile producers do not mention their will to adopt these initiatives.

**Climatic hazard.** Facing feed shortages due to unfavourable climatic conditions, producers tending toward GBI and LBE seem not to have the same way of thinking; GBI producers

**Table 7. Value and level of significance for the difference in the ideal future farm gradient as a function of modalities of categorical variables: Environmental aspects (N = 207).**

| Categorical variable | Modality and estimate | | *P* |
|---|---|---|---|
| **Considerations of environmental aspects** | | | |
| Measurement of the grass height: optimisation practice | Yes | No | 0.059 |
| | −0.27 | 0.0083 | |
| Forage mixture with leguminous plants: optimisation practice | Yes | No | 0.0088 |
| | −0.11 | 0.083 | |
| Field notebook: optimisation practice | Yes | No | 0.065 |
| | −0.074 | 0.061 | |
| **Climatic hazard** | | | |
| Increase of concentrate distribution: strategy to confront climatic hazards | Yes | No | 0.036 |
| | 0.22 | −0.036 | |
| Decrease of the herd: strategy to confront climatic hazards | Yes | No | 0.037 |
| | −0.25 | 0.014 | |
| Food self-sufficiency: cause for maintaining constant or decreased milk production | Yes | No | 0.14 |
| | −0.17 | 0.0073 | |

Means with different letters are significantly different.

intend to buy high nutritional feed to balance shortages (estimate = 0.22, Table 7) and LBE producers are going to decrease the number of cows (estimate = −0.25, Table 7) and ensure their feed autonomy (estimate = −0.17, Table 7).

### How do farmers' ideal future farm compare to their current farming systems?

The current situation of dairy producers was compared to their preferred IFF (Table 1). Except for the type of workforce, quite high percentages of "unhappy" producers were observed for the farm characteristics, between 37 to 50%. This suggested that not all producers work as they would like to. The same comparison was not found in the literature, to our knowledge.

As dairy producers do not work in a way that they consider to be ideal, it is interesting to study the gaps to fill in order to reach their ideal system and so, amongst others, their needs. The study of the requirements to reach the IFF, including ways to meet these needs and the area of the needs, can inform the stakeholders of the dairy sector about what must be developed to evolve into IFF.

### Which paths and themes of training do dairy producers want in order to reach their desired ideal future farm?

**Paths to formation.** As way to improve their skills, producers wanting GBI tended to favour consultancy (estimate = 0.17, Table 8) and commercial companies (estimate = 0.16, Table 8) and not days of study on other farms (estimate = 0.082, Table 8), meanwhile producers wanting LBE supported this latter possibility (estimate = −0.088, Table 8), a network of pilot farms (estimate = −0.13, Table 8) and the associate, non-market sector (estimate = −0.21, Table 8). Moreover, for help in technical choices, producers desiring LBE chose formation and study days (estimate = −0.15, Table 8) and producers' technical groups to implement in the future (estimate = −0.20, Table 8). The choices presented confirm the will for a non-market way to learn for producers wanting LBE, contrary to producers wishing for GBI.

As an information source, the agricultural press was commonly cited (N = 161, *i.e.* 78% of respondents), but producers desiring LBE tend to not want to inform themselves in this conventional way (estimate = −0.14, Table 8).

Producers wanting a GBI model tend to need more help to free them from their work in order to follow a formation (R = 0.21, Table 6)

**Formation domains.** The formation domains reflected the direction chosen by producers looking for LBE and the ways to reach it. They tend to want skills related to processing and diversification (estimate = −0.18, Table 8) and were likely to reject finance, management (estimate = −0.24, Table 8), administrative (estimate = −0.11, Table 8) and legal framework skills (estimate = −0.083, Table 8). For financial aspects producers wanting LBE tend to favour requests for advice from experts rather than self-formation (estimate = −0.15, $P_{\text{to implement } vs. \text{ not interested}}$ = 0.12, Table 8). They do not choose animal feeding (estimate = −0.14, Table 8) and selection formations (estimates = −0.053; −0.082, Table 8). This could suggest the will of the producers not to change their way of management and the level of quality of their herd but the method of valorisation of their production.

In contrast, producers desiring GBI tend to want to continue to enhance their vegetal and animal production (estimates = 0.083; 0.08, Table 8), to become more efficient and enhance their revenue. Moreover they are more interested in legal aspects (estimate = 0.14, Table 8). Expansion and complexification of the GBI model of dairy farms wished for by these producers could be an explanation. Bergevoet *et al.* [12] also observed a will to be well informed about the legislation for the "modern and large farm" profile. This is not noted in their profile, which is close to the LBE model.

**Table 8. Value and level of significance of the difference in the ideal future farm gradient as a function of modalities of categorical variables: Formations (N = 207).**

| Categorical variable | Modality and estimate | | | P |
|---|---|---|---|---|
| | **Needs: ways to learn formations** | | | |
| **Consultancy company: SFI* place | Yes | | No: | 0.0017 |
| | 0.17 | | −0.087 | |
| **Study days on farm: SFI place | Yes | | No | 0.026 |
| | −0.088 | | 0.082 | |
| **Network of pilot farms: SFI place | Yes | | No | 0.025 |
| | −0.13 | | 0.056 | |
| **Associate, non-market sector SFI place development | Yes | | No | 0.0023 |
| | −0.21 | | 0.062 | |
| **Commercial company: SFI place | Yes | | No | 0.014 |
| | 0.16 | | −0.058 | |
| ***Agricultural press: information source | Yes | | No | 0.068 |
| | 0.025 | | −0.14 | |
| Formation and study day: help for technical choices | Already implemented 0.037[a] | To implement in the future−0.15[b] | Not interested 0.035[a] | 0.082 |
| Producers technical groups: help for technical choices | Already implemented 0.020[a] | To implement in the future−0.20[b] | Not interested 0.11[a] | 0.0046 |
| | **Needs: domain formation** | | | |
| **Finance and management: requested formation | Yes | | No | 0.0007 |
| | 0.066 | | −0.24 | |
| **Processing and diversification: requested formation | Yes | | No | 0.0008 |
| | −0.18 | | 0.089 | |
| **Plant selection: requested formation | Yes | | No | 0.087 |
| | 0.083 | | −0.053 | |
| **Animal selection: requested formation | Yes | | No | 0.034 |
| | 0.080 | | −0.082 | |
| **Animal feeding: requested formation | Yes | | No | 0.073 |
| | 0.03 | | −0.14 | |
| **Administrative: requested formation | Yes | | No | 0.026 |
| | 0.064 | | −0.11 | |
| **Legal framework: requested formation | Yes | | No | 0.005 |
| | 0.14 | | −0.083 | |
| Request for advice: help for financial aspects | Already implemented 0.014[a] | To implement in the future−0.15[b] | Not interested 0.049[a] | 0.19 |

Means with different letters are significantly different.

*SFI = study, formation and information.

**producers declaring no will of formation were removed from this analysis.

*** producers declaring no agricultural press as an information source were removed from this analysis.

Two kinds of formation were identified and preferred by producers wanting LBE or GBI models. Bergevoet *et al.* [12] observed the will to innovate for the two profiles closest to LBE and GBI profiles of this study. Verhees *et al.* [15] observed that formation was the most important resource for dairy producers. The present research differentiated the formation desired as a function of IFF. Dufour *et al.* [43] defined, through a survey of 15 dairy farmers, three

conceptions of the work: difficult, organisational and passionate. The passionate approach was accompanied by the desire for new knowledge which was, as observed here, either to learn about genetic selection or about processing and marketing of products.

## Conclusions

In conclusion, the GBI tendency is two times more represented than the LBE tendency. Many reasons explain this choice of ideal farm. Past crises seem to cause farmers to desire the LBE model. A high workload seems to orientate respondents to the GBI model. The wish for the IFF is influenced by the current framework but is also a question of mentality. Production factors reached, breeds chosen for the herd, ways to react to factors external to the farm, consideration of diversification and alternative valorisation, regrouping and mechanisation and robotisation describe the producers' mentality and showed different relations with the IFF chosen. Moreover LBE and GBI producers may both have concern for the environment, but the approach to act for the environment by LBE producers, through concern for ecosystem services, is clearly highlighted in this study. These producers found it important to answer to society's expectations. Finally, as the current situation of farming is quite different to the ideal one, the learning needs were studied and two types of customer appeared in relation to their formation. We conclude that two kinds of dairy producers seem to appear, for different reasons, with different relations to the environment and asking for different formations. The appearance of these two opposite tendencies in an agricultural sector were observed in a highly heterogeneous region with regard to soil and geological characteristics and so could be observed in similar contexts of production in other countries.

## Supporting information

**S1 Appendix. Questions of the survey mobilized in the study.**
(DOCX)

## Acknowledgments

I want to thank the organising committee of "Carrefour des Productions animales" for the supply of the data.

## Author Contributions

**Conceptualization:** Anne-Catherine Dalcq, Thomas Dogot, Yves Beckers, Hélène Soyeurt.

**Data curation:** Anne-Catherine Dalcq, Frédéric Vanwindekens, Hélène Soyeurt.

**Formal analysis:** Anne-Catherine Dalcq, Eric Froidmont, Frédéric Vanwindekens.

**Investigation:** Anne-Catherine Dalcq, Eric Froidmont.

**Methodology:** Anne-Catherine Dalcq, Thomas Dogot, Yves Brostaux, Hélène Soyeurt.

**Software:** Anne-Catherine Dalcq, Yves Brostaux, Hélène Soyeurt.

**Validation:** Anne-Catherine Dalcq.

**Writing – original draft:** Anne-Catherine Dalcq.

**Writing – review & editing:** Anne-Catherine Dalcq, Thomas Dogot, Yves Beckers, Yves Brostaux, Eric Froidmont, Frédéric Vanwindekens, Hélène Soyeurt.

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
