## [Decision Letter · Decision Letter 0]

6 Feb 2020

PONE-D-19-26278

How do dairy farmers wish their future farm?

PLOS ONE

Dear Miss Dalcq,

Thank you for submitting your manuscript to PLOS ONE. After careful consideration, we feel that it has merit but does not fully meet PLOS ONE’s publication criteria as it currently stands. Therefore, we invite you to submit a revised version of the manuscript that addresses the points raised during the review process:

More precisely : 

- Please Improve the introduction (see reviewer 1)

- You have to clarify the methodology of your survey (reviewers 1 and 2)

- Justify the statistical analysis: the reviewers (2 and 3) propose two different possible extensions (Latent Class Analysis or, Hierarchical Cluster Analysis)  that are in line with your purpose (a typology of farms) 

- Please have a look at the additional literature suggested by reviewer 2

In addition to the reviews, you will find also as an attached file the comments made by reviewer 2 on your manuscript.

We would appreciate receiving your revised manuscript by Mar 22 2020 11:59PM. To enhance the reproducibility of your results, we recommend that if applicable you deposit your laboratory protocols in protocols.io, where a protocol can be assigned its own identifier (DOI) such that it can be cited independently in the future. For instructions see: http://journals.plos.org/plosone/s/submission-guidelines#loc-laboratory-protocols

We look forward to receiving your revised manuscript.

Kind regards,

Damien Rousselière, PhD

Academic Editor

PLOS ONE

Journal Requirements:

5. Your ethics statement must appear in the Methods section of your manuscript. If your ethics statement is written in any section besides the Methods, please move it to the Methods section and delete it from any other section. Please also ensure that your ethics statement is included in your manuscript, as the ethics section of your online submission will not be published alongside your manuscript.

Reviewers' comments:

Reviewer's Responses to Questions

**Comments to the Author**

1. Is the manuscript technically sound, and do the data support the conclusions?

Reviewer #1: Partly

Reviewer #2: Partly

Reviewer #3: Yes

2. Has the statistical analysis been performed appropriately and rigorously? 

Reviewer #1: I Don't Know

Reviewer #2: Yes

Reviewer #3: Yes

3. Have the authors made all data underlying the findings in their manuscript fully available?

Reviewer #1: No

Reviewer #2: No

Reviewer #3: No

4. Is the manuscript presented in an intelligible fashion and written in standard English?

Reviewer #1: No

Reviewer #2: Yes

Reviewer #3: Yes

5. Review Comments to the Author

Reviewer #1: Dear authors,

The theme of the paper is very interesting but in my point of view the paper is not good enough at the present moment to be published in this journal. I believe you need more time to improve the manuscript for several reasons. The first is the English. Many parts of the text I could not understand what you want to say including wrong words. Also, I believe that you not formatted the text as the journal guidelines – for example the paper should be on .doc format, not pdf. The first page with the items is absolutely unnecessary and anyone include this to submit any paper. Also, the archive missed the number pages after the first table, and the Figure 1 is not included. Is impossible to evaluate your results without it. Finally, I am not aware about any good journal that publish results and discussion of a study together. I strongly recommend to separate this section on two sections. You can note that you only comment some results rather than really discuss them. This is why we separate the sections.

Another general comments:

Introduction: The introduction is too brief in the literature review. You explain some characteristics that actually would be in the methodology section. I believe the introduction need an improvement showing us other studies to better contextualize the problem because is not clear enough. Also you say in the introduction that the study was performed just before the end of European milk quota. Do you think this fact impact in your data? How? Please find some place to write about this fact and the impact in your study.

Methodology: The methodology is confused. We do not know for example how the survey was lauched but we have the information about how many farmers were surveyed, that actually could be said in the results. Please start the methodology from the begining of your project.

Some specific comments:

Ln 84: Who “asked”? Please contextualize.

Ln 90-91: It is important to say what they found on these researches.

Ln 91: “realized”: I think this word is in the wrong place... did you mean "performed"?

Ln 93: Which choices? You did show us which paths they found. The last sentence says nothing.

Ln 94: You did not present those questions so is bad to say that your study is different. Readers are not aware of these studies.

Ln 101: Which choice?

Ln 105: “impacts of agriculture and breeding on the environment” Such as…?

Ln 105: Which "formation aspect"? What do you mean? I did not understand this entire sentence…

Ln 110: “was” instead “are”.

Ln 112 – 117: This explanation is not necessary here, is like a summary and here is the introduction.

Ln 120 – 121: Usually we say that in the end of the methodology when we are describing the analysis... it is not good here. Please start the methodology from the begining.

Ln 123: But where the study was carried out?

Ln 123: "a survey"? or "this/our survey"? It means that is another survey that not yours.

Ln 124 – 125: Why you emphasize this but do not explain the implications of this?

Ln 131 – 132: Bring this to the begining. Again, start from the begining.

Ln 133 – 134: These are results and this paragraph has only one sentence.

Ln 177: Table 1 is showing results so should not be cited here.

After the Table 1 the archive lost the page numbers so I’m not able to do my comments. Also, where the Figure 1? I can not evaluate what you saying without it.

Reviewer #2: This study examines dairy farmers' preferred future for their farms, how close their farm is to this ideal, the reasons they arrived at this ideal, and the environmental aspects of the ideal. Using a survey of producers, the author(s) conduct MCA to create a gradient of "ideal future farm" ranging from local-extensive (LBE) to global-intensive (GBI). They then conduct Pearson correlations between this gradient and several key variables (including environmental aspects and decision-making) and finally compare the gradient score (preference) to the producers' current farm characteristics to assess whether the producers have a current farm that fits with their stated ideals.

It is noted that in the memo and manuscript the authors make no mention of having received ethics approval from their institutions for the survey used to collect data from dairy farmers. This must be addressed through correspondence with the editors prior to publication. In the United States, research such as this survey would have needed to receive prior approval/authorization from an Institutional Review Board. I'm not clear what the equivalent would be within the authors countries, but the editors should confirm that appropriate ethical standards were followed including voluntary participation and confidentiality. It appears that the authors have attached some type of letter of confirmation from their home university, which should be reviewed by the editors.

The study is quite interesting and could make a valuable contribution with some important changes and improvements. The statistical analyses seem to be used appropriately throughout and the findings are quite interesting. In particular, the authors' discussion of how farmers arrive at their IFF and the relationship to environmental factors is enlightening. With revisions, the paper could have quite an impact. I raise my concerns for revision below in order of priority.

Firstly, the authors must add much more methodological detail to their discussion of the survey. Specifically, they must address how the sample was obtained, generalizability, how the survey was actually administered (mail? online? etc.) and why that was appropriate, response rate (and non-response analysis), pre-testing, etc. They should also discuss how missing data and "no opinion" was coded and treated in the MCA and GLM.

Secondly, it is not clear to me why the authors use MCA instead of Latent Class Analysis. The goals of the study - to group farmers/respondents into classes - and the language that the authors use to discuss the MCA results feels more appropriate to LCA. Is there a specific reason they did not use LCA? If not, I would encourage them to consider using LCA or at least running LCA as a robustness check.

Thirdly, it is not always clear how the concepts were actually measured in the survey. Some of the core concepts are quite complex, and the authors do not tell us what they actually asked the farmers and how they got from those survey questions to the concepts that they coded. Either a link to the survey text somewhere online or an appendix to provide more detail on the actual measures is necessary. Things like intensive/extensive, environmental aspects, are complex concepts and it needs to be clear how they were actually measured and why they were coded as simple binary variables.

There are two relevant rural sociology literatures that the authors should explore: one regarding the contradictory class location of family farms and one regarding bifurcation of farm systems. The first should start with the book My Own Boss by Pat Mooney and those who cite him. Mooney discusses how farmers exist as both workers and capitalists (employ others) and this literature would complicate the family/employee line the authors are trying to draw. The second literature describes the divide occurring in agriculture between large-intensive farms and small-local farms, calling this process bifurcation. This would bolster the authors' argument to engage with this literature.

Regarding the writing, I would encourage the authors to revise the introduction to set up their research using questions rather than statements. This would make it much more clear to readers what the aim is of their study. I have provided some specific notes and suggestions in the PDF. Also, in the results section, there is much that should be removed and moved to the discussion and conclusion. The comparisons to other studies should be saved for the discussion and conclusion, not results. It makes it difficult to grasp your key findings (particularly when so much is covered) when you are constantly delving into other literature. That literature should be saved for a discussion section.

It is a small, but important, note: the authors repeatedly use the term "qualitative" to describe some of their variables. These measures are not qualitative, they are categorical or nominal. Within social science research, qualitative has a specific meaning (meaning data derived from text, interviews, observation, etc.) and they should avoid using that term when that's not really what their data entails. Throughout the manuscript, this should be changed to "categorical", "nominal", or "binary" when appropriate.

Reviewer #3: The paper is clear and pleasant to read. Reporting the results of a survey on the ideal future farm for Walloon dairy farmers, this empirical research sheds light on the (ex ante) attitudes of farmers.

The methods used are simple and do not add value, but they are clearly explained and used correctly.

The results of the analysis are clearly explained. I would like to stress that a major effort has been made to compare the results with those of other neighbouring areas (France etc.). The comparison is of particular interest to the reader, as it allows us to situate this research work in relation to the existing literature, but also to underline the originality of the work and the new knowledge provided by this applied work on the aspirations of dairy farmers in Wallonia .

A few remarks:

- The analysis of the survey focuses on farmers' responses to questions regarding their perception of the ideal future farm. On page 13, it is written that "no opinion" farmers are excluded from the analysis to avoid potential bias. However, I would have liked to have seen a specific paragraph in which these "no opinion" farmers are analysed, even if only briefly. For example, is the answer 'no opinion' characteristic of a particular type of farmer? Furthermore, the proportion of these respondents is itself an indication of the lack of perspective that some dairy farmers may feel. The analysis of this sub-sample can therefore complement the overall analysis.

- The MCA is carried out correctly, and the conclusions you draw show that there are two main types of farmers, both in terms of their current activity and their vision of the ideal future farm. It would have been interesting, and it is advisable, that you supplement your analysis with a hierarchical clustering, whether divisive or agglomerative. Such an analysis will provide a solid complement to your data processing and could corroborate and even support your conclusions.

- The main conclusions of the analysis highlight dairy farmers' vision of the future of their sector. As written in the text, the survey dates from 2015. Is it possible to compare the results with more recent statistics on the development of dairy activity in Wallonia? In other words, are there data to make a comparison between ex ante attitudes and ex post behaviours?

6. PLOS authors have the option to publish the peer review history of their article (what does this mean?). If published, this will include your full peer review and any attached files.

Reviewer #1: No

Reviewer #2: No

Reviewer #3: No

---

## [Author Response · Author response to Decision Letter 0]

15 Apr 2020

All the answers to the reviewers and the academic editor are also present in the document 'Response to reviewers'.

Thank for your remarks which improve my paper.

Reviewer #1: Dear authors,

The theme of the paper is very interesting but in my point of view the paper is not good enough at the present moment to be published in this journal. I believe you need more time to improve the manuscript for several reasons. The first is the English. Many parts of the text I could not understand what you want to say including wrong words. 

A-C D : The English was revised by a native speaker, Cate Chapman, which is a professional to review the English of scientific papers. The modifications made on the manuscript following your remarks were also revised by Cate Chapman.

Also, I believe that you not formatted the text as the journal guidelines – for example the paper should be on .doc format, not pdf. 

A-C D : I don’t understand this comment as the article that I submitted was a .docx document, not a pdf.

The first page with the items is absolutely unnecessary and anyone include this to submit any paper. 

A-C D : You are right. I included a table of contents as I understood this demand of PLOSONE in the guidelines but when reading these ones again, I think I made a mistake. This table is not deleted.

Also, the archive missed the number pages after the first table, and the Figure 1 is not included. Is impossible to evaluate your results without it. 

A-C D : I admitted I made a mistake by forgetting the line numbers after the first table but not the page numbers. I corrected this in the new version. I don’t understand the problem with the figure as it was included when I submitted the article.

Finally, I am not aware about any good journal that publish results and discussion of a study together. I strongly recommend to separate this section on two sections. You can note that you only comment some results rather than really discuss them. This is why we separate the sections.

A-C D : PlosOne guidelines allowed the organization of the manuscript using a common Results and Discussion section. According to me, it was more suitable for the current article. It allows a progressive presentation of the results and their discussion directly step by step. We have also improved the clarity of this section using questions as proposed by one reviewer.

See the PlosOne below:

“Results, Discussion, Conclusions

These sections may all be separate, or may be combined to create a mixed Results/Discussion section (commonly labeled “Results and Discussion”) or a mixed Discussion/Conclusions section (commonly labeled “Discussion”). “

Another general comments:

Introduction: The introduction is too brief in the literature review. I believe the introduction need an improvement showing us other studies to better contextualize the problem because is not clear enough. 

A-C D : I have developed what authors found through their researches in the discussion, when I have compared my results to them. In order to limit the repetitions, I have decided to limit the references in the introduction section. Consequently, in the introduction section, I have warned that some authors have already observed trends and highlighted the innovative aspects related to our contribution. However, according to your comment, I have modified some sentences to make them clearer (lines 66-75). Moreover, another reviewer asks me to shorten my introduction by putting more literature review in the discussion. So this also explained my current position. 

Introduction: You explain some characteristics that actually would be in the methodology section.

A-C D : Could you mention the lines in order to help my understanding? Indeed, I don’t know where is the problematic section.

Also you say in the introduction that the study was performed just before the end of European milk quota. Do you think this fact impact in your data? How? Please find some place to write about this fact and the impact in your study.

A-C D : As proposed by you, I have completed my idea at the lines 81-85 of this new version. Thanks for your remark, it made my article clearer.

Methodology: The methodology is confused. We do not know for example how the survey was lauched but we have the information about how many farmers were surveyed, that actually could be said in the results. Please start the methodology from the begining of your project.

A-C D : You are right. The methodology about the survey protocol was not enough detailed. All information is now added in the Material and Methods section (lines 109-120). Moreover a new section entitled “Data representativeness” was created in this new version to compile all information regarding the surveyed dairy farmer’s population (lines 178-193). 

Some specific comments:

Ln 84: Who “asked”? Please contextualize.

A-C D : It is now precised at lines 64 and 65.

Ln 90-91: It is important to say what they found on these researches.

A-C D : As mentioned in my previous answer to this similar, I have developed with more details what authors found through their researches in the discussion section in order to confront my results. This allows to limit the redundancies throughout the manuscript.

Ln 91: “realized”: I think this word is in the wrong place... did you mean "performed"?

A-C D : You are right, “performed” suits well in this sentence. The change was done at line 74.

Ln 93: Which choices? You did show us which paths they found. The last sentence says nothing.

A-C D : You are right. This sentence was modified at line 75.

Ln 94: You did not present those questions so is bad to say that your study is different. Readers are not aware of these studies.

A-C D : The sentence was modified (lines 77-78). I have never seen this question in other studies but now in the manuscript, I have precised that it is to our knowledge.

Ln 101: Which choice?

A-C D : The choice is now better mentioned (line 87). 

Ln 105: “impacts of agriculture and breeding on the environment” Such as…?

A-C D : The sentence was modified (line 91-92). 

Ln 105: Which "formation aspect"? What do you mean? I did not understand this entire sentence…

A-C D : The formation aspect concerns the training aspect. The sentence was modified (line 89). 

Ln 110: “was” instead “are”.

A-C D : When I spoke about the objectives of my research, I use present tense as these goals are untemporal and multiple.

Ln 112 – 117: This explanation is not necessary here, is like a summary and here is the introduction.

A-C D : I think this statement of objectives are important at the end of introduction to refocus the objectives at the end of the introduction. However, as proposed by the second reviewer, the objectives are now presented throughout questions. This introduction part is now “lighter” as asked by you.

Ln 120 – 121: Usually we say that in the end of the methodology when we are describing the analysis... it is not good here. Please start the methodology from the beginning.

A-C D : I moved this sentence at the end of the material and methods section (Lines 169-170). 

Ln 123: But where the study was carried out?

A-C D : : New sentences were added to bring that clearer (lines 109-120).

Ln 123: "a survey"? or "this/our survey"? It means that is another survey that not yours.

A-C D : It is now precised at line 121.

Ln 124 – 125: Why you emphasize this but do not explain the implications of this?

A-C D : As mentioned previously, I have now completed my idea by adding sentences (lines 81-85).

Ln 131 – 132: Bring this to the begining. Again, start from the begining.

A-C D : I have changed the beginning of the Material and methods section accordingly (lines 109-120 & lines 184-185).

Ln 133 – 134: These are results and this paragraph has only one sentence.

A-C D : As proposed, it was moved in the Results and Discussion section (lines 186-187). 

Ln 177: Table 1 is showing results so should not be cited here.

A-C D : This reference was done to illustrate the method thanks to the titles of lines and columns of the table. It is to improve the understanding of the method. As no other reviewer made this remark, we have chosen to let this illustration.

After the Table 1 the archive lost the page numbers so I’m not able to do my comments. Also, where the Figure 1? I can not evaluate what you saying without it.

A-C D : I don’t understand the problem as Figure 1 was well submitted. I did not check if the line numbers continue after the first section break. Sorry for this mistake.

Reviewer #2: This study examines dairy farmers' preferred future for their farms, how close their farm is to this ideal, the reasons they arrived at this ideal, and the environmental aspects of the ideal. Using a survey of producers, the author(s) conduct MCA to create a gradient of "ideal future farm" ranging from local-extensive (LBE) to global-intensive (GBI). They then conduct Pearson correlations between this gradient and several key variables (including environmental aspects and decision-making) and finally compare the gradient score (preference) to the producers' current farm characteristics to assess whether the producers have a current farm that fits with their stated ideals.

It is noted that in the memo and manuscript the authors make no mention of having received ethics approval from their institutions for the survey used to collect data from dairy farmers. This must be addressed through correspondence with the editors prior to publication. In the United States, research such as this survey would have needed to receive prior approval/authorization from an Institutional Review Board. I'm not clear what the equivalent would be within the authors countries, but the editors should confirm that appropriate ethical standards were followed including voluntary participation and confidentiality. It appears that the authors have attached some type of letter of confirmation from their home university, which should be reviewed by the editors.

A-C D : In Belgium, five years ago, no agreement from Ethics committee was needed. We have given to the editor a letter of the committee confirming this fact.

The study is quite interesting and could make a valuable contribution with some important changes and improvements. The statistical analyses seem to be used appropriately throughout and the findings are quite interesting. In particular, the authors' discussion of how farmers arrive at their IFF and the relationship to environmental factors is enlightening. With revisions, the paper could have quite an impact. I raise my concerns for revision below in order of priority.

A-C D : Thanks for your opinion about the qualities of my paper.

Firstly, the authors must add much more methodological detail to their discussion of the survey. Specifically, they must address how the sample was obtained, generalizability, how the survey was actually administered (mail? online? etc.) and why that was appropriate, response rate (and non-response analysis), pre-testing, etc. 

A-C D : You are right. This information was lacking. More details are now mentioned in the new manuscript (lines 109-120). 

They should also discuss how missing data and "no opinion" was coded and treated in the MCA and GLM.

A-C D : Only surveys completely answered were analysed in this research. Therefore, we had no missing data. This is now precised (line 121). Each question proposed the two contrary choices and the No opinion proposal. As explained at lines 144-145, cluster analysis was used to exclude the group of producers with particular characteristics. This was done to remove the producers with a No-opinion profile (cf Figure 1 and lines 235-237). So, these producers were not taken into account for the GLM and Pearson correlation analyses.

Secondly, it is not clear to me why the authors use MCA instead of Latent Class Analysis. The goals of the study - to group farmers/respondents into classes - and the language that the authors use to discuss the MCA results feels more appropriate to LCA. Is there a specific reason they did not use LCA? If not, I would encourage them to consider using LCA or at least running LCA as a robustness check.

A-C D : The first goal of the article was to study the relationships between the modalities of the seven IFF sub-questions. Some sentences were added to better highlight this point (lines 133-134). Latent class analysis does not allow that. The idea was not to force the creation of groups of producers. Then two gatherings of modalities were observed. But, there was a will to not put producers into a pigeonhole but speak about the producers to have a tendency to be for a model or another. Some information was now added in the text to improve the understanding (lines 348-349).

Thirdly, it is not always clear how the concepts were actually measured in the survey. Some of the core concepts are quite complex, and the authors do not tell us what they actually asked the farmers and how they got from those survey questions to the concepts that they coded. Either a link to the survey text somewhere online or an appendix to provide more detail on the actual measures is necessary. Things like intensive/extensive, environmental aspects, are complex concepts and it needs to be clear how they were actually measured and why they were coded as simple binary variables.

A-C D : I put more details about the survey and its internet link in the Material and Methods section (lines 109-132). Intensive/ extensive were proposed without other explanations. Even considering bias, it give an insight of what is the tendency of the producer regarding this question.

There are two relevant rural sociology literatures that the authors should explore: one regarding the contradictory class location of family farms and one regarding bifurcation of farm systems. The first should start with the book My Own Boss by Pat Mooney and those who cite him. Mooney discusses how farmers exist as both workers and capitalists (employ others) and this literature would complicate the family/employee line the authors are trying to draw. The second literature describes the divide occurring in agriculture between large-intensive farms and small-local farms, calling this process bifurcation. This would bolster the authors' argument to engage with this literature.

A-C D : Thanks to inform me about these researches. I have explored these two literatures and implemented the contents of those article in this new manuscript (lines 215-217; lines 338-341; lines 538-541). Concerning bifurcation, I did not find articles about bifurcation between large-intensive farms and small-local farms as you said but I read phenomenon of bifurcation for the organic sector (lines 338-341). 

Regarding the writing, I would encourage the authors to revise the introduction to set up their research using questions rather than statements. This would make it much more clear to readers what the aim is of their study. I have provided some specific notes and suggestions in the PDF. 

A-C D : I have changed the end of my introduction to present the goals of my research using research questions (lines 97-104). Thanks for this suggestion and your proposals, it made the article topic clearer. For more clarity, I have also changed the titles of the different parts in the Results and discussion section using these same research questions.

Also, in the results section, there is much that should be removed and moved to the discussion and conclusion. The comparisons to other studies should be saved for the discussion and conclusion, not results. It makes it difficult to grasp your key findings (particularly when so much is covered) when you are constantly delving into other literature. That literature should be saved for a discussion section.

A-C D : We have decided to write a common Results and Discussion section because according to me, it was more suitable for my article. Indeed, this allows a progressive presentation of the results and their discussion directly step by step. I preferred to address each theme one by one as they are quite different but they followed a logic stream. However, I have changed the titles of this part based on the formulated research questions as you proposed in your past comments to improve its clarity.

It is a small, but important, note: the authors repeatedly use the term "qualitative" to describe some of their variables. These measures are not qualitative, they are categorical or nominal. Within social science research, qualitative has a specific meaning (meaning data derived from text, interviews, observation, etc.) and they should avoid using that term when that's not really what their data entails. Throughout the manuscript, this should be changed to "categorical", "nominal", or "binary" when appropriate.

A-C D : Thank you for this comment. I used “qualitative” as a contrary of “quantitative”. I was not aware that in social science research it could have another meaning. In the current manuscript, the term “qualitative” was replaced by “categorical”.

Abstract : MCA is like PCA but for binary variables. The only potential concern is the overly simplistic representation of the concepts with binaries.

A-C D : MCA shares a similar concept with PCA about the dimensionality reduction. However MCA is based on frequencies of modalities. However, it is not true to mention that MCA is for binary variables. Categorical variables can be also used in this kind of method reducing the dimensionality of X matrix. MCA was appropriate to this study as we don’t want to group farmers but we want to observe their tendency to belong to one or another group. It is why we have used the concept of gradient.

Abstract : The only concept here that's not immediately interpretable by me is the intensive versus extensive

A-C D : As mentioned in a previous comment, Intensive or extensive were proposed without other explanations. Even considering bias, it gave an insight of what is the tendency of the producer regarding this question.

Ln 61 : These two sentences do nothing to build the study up and are inflammatory (and probably inaccurate). 

A-C D : We consider that these two sentences have sense because it reminds the sense of the profession of producer. It is a profession amongst others but which provides a basic need of the society. And this profession is at a turning point with its separation between two models at our time, as observed in the article. I do not think it is inflammatory, I added references to objectify this phenomenon (line 43).

Ln 69 : Quite a bit of this background here before getting to your actual questions. I would encourage you to frame your questions and study earlier.

A-C D : I am not sure to have understood your comment. However, the current manuscript is now reformatted based on research questions as proposed in one of your previous comment. The structure is now clearer. 

Ln 95 : So the ideal type is only revenue specific? Or does it encompass other goals for the farmer and farm?

A-C D : The question was asked following this idea. It is now precised in the manuscript (lines 124-126). 

Ln 101 : The wording here is not clear. "This choice" meaning the choice of an ideal? Be more specific in the wording.

A-C D : This sentence was rewritten (line 87).

Overall, the abstract does a much better job of setting up the study than this introduction is doing. What are your questions? Stating them as research questions will make it much more clear.

A-C D : As mentioned previously, it is now the case. Thank you for this proposal.

Ln 103 : Not clear what is meant by "formational"

A-C D : This sentence was rewritten (line 89).

Ln 109-112 : This is much more clear. Again, stating the research questions as QUESTIONS would help. What are dairy farmers "ideal future farming systems" (IFF)? How do farmers decide on their IFF? How do environmental aspects factor into IFF decisions? And, ultimately, how do farmers' IFF compare to their current farming systems?

A-C D : You are right. The manuscript is now re-structured using research questions. This improve greatly its clarity.

Introduction : Break the introduction and the literature review into discrete sections to allow your questions to be more clear, your case set up (this comes in introduction) and then give more depth on the studies that you are reviewing. 

A-C D : The introduction was voluntary short to limit the redundancies with the discussion section. I have developed in the details the past studies in the discussion section in order to confront my results. However, now the end of this introduction was revised and the objectives were mentioned using research questions (lines 97-104) as proposed by you in one of your previous comments.

Ln 123 : Discuss the representativeness of the sample. From this group, can you generalize to the Belgian producers generally? What was the sampling approach? Was it random? What was the response rate to the survey, and did they conduct any non-response analysis to confirm that there is no bias? Ln 126 : They must describe this methodology in much greater detail. How was the sample obtained? Response rate? How was the survey administered? Was it pre-tested? Who on the farm actually filled out the survey? All of these things must be discussed. 

A-C D : You are right. The description was not enough complete. Information were now added in the manuscript (Lines 109-122). 

Ln 136 : They are all quantitative, it's just that some are categorical.

A-C D : Among all variables present in the survey, some of them were not quantitative. They measured not a quantity but inform on a quality.

Ln 142 : What about other types of missing data, such as skipped questions? How was this dealt with?

A-C D : We had no problem with missing data as only surveys fully completed were kept for this analysis. It is precised in the manuscript at line 121. 

Ln 142 : How was this coded? Were these responses dropped out of the MCA and GLM?

A-C D : Details are given at lines 133-135 & 235-239.

Ln 147 : Again, not qualitative: categorical (or nominal). Qualitative has a specific meaning, and you should not use it this way.

A-C D : As mentioned previously based on your previous comments, the term “qualitative” was replaced by “categorical”.

Ln 150 : Did the authors explore using latent class analysis? Why or why not? It seems that LCA might be more appropriate than MCA for what the authors are trying to do which is classify respondents into categories.

A-C D : See the previous answers to your comments about the comparison between MCA and PCA as well as the interest of using MCA to define the gradient.

Ln 152 : If you want to use the term "classes" to discuss the grouping of the respondents, than Latent Class Analysis seems much more appropriate than MCA.

A-C D : Almost 95% of the variability is explained by the two first dimensions. The first separation of the cluster analysis was between the No-opinion and the other producers, which permitted to identify and isolate them. The second separation would be along the second dimension, the creation of classes encountered this objective. As mentioned previously, the latent class analysis was not suitable for our study.

Ln 166 : How did you determine which variables to test the relationship with?

A-C D : We have tested the relationships with all the variables related to the chosen themes (i.e., reasons, environmental aspects and training). Only variable with significant relationships with the IFF gradient are presented in this article. However, we have decided to present the results about two variables (i.e., age and geographical situation) which were not significant because we found them interesting for the interpretation. 

Ln 180 : It would be helpful to provide a link to the actual survey text (perhaps translated) or more detail on the actual questions asked that became the core concepts. For instance, how were farmers asked about "intensive" versus "extensive"? These are complex concepts and not everyone would agree on the same meaning.

A-C D : The internet link of the survey is now added in the Material and Methods section (line 120). Intensive/ extensive were proposed without other explanations. Even considering bias, it give an insight of what is the tendency of the producer regarding this question.

Again, need to provide us with information about how these were actually measured. What did you actually ask farmers to get to these concepts? Why are these complex concepts represented only as binary variables? Particularly for questions about someone's IDEAL, you need to tell us what you actually asked them

A-C D : The link of the survey is now available at line 120. Sentences were added to precise the way of these questions was proposed (lines 124-131).

This especially seems problematic. Don't most farms use combinations of both family and non-family? Is the family meaning NO employed labor? This is why you need to give us more detail, probably as an[…] [the sentence is not complete in the comment]

A-C D : The questions were voluntary sliced to have their point of view (more a system or another?) in order to know what they consider as ideal.

There is a related rural sociology literature on the fact that family farmers are BOTH workers and employ other workers. You should look for pieces by Mooney (My Own Boss) and others who cite him

A-C D : Thank you for this suggestion. I have explored this literature and implemented the contents of this article in the new manuscript (lines 215-217; lines 538-541).

It is strange to mix the review of other studies into the results section here. This is all discussion and conclusion, not results.

A-C D : As mentioned previously, we have decided to write a common Results and Discussion section because according to us, it was more suitable for this article. It allows a progressive presentation of the results and their discussion directly step by step. I preferred to address each theme one by one as they are quite different but they followed a logic stream. However, the structure is now clearer thanks to your suggestion to use research questions.

The authors should look at the BIFURCATION literature and use that term.

A-C D : As mentioned previously, I have explored this literature and implemented the content of these articles in the manuscript (lines 338-341).

This table is quite hard to read as a whole. Perhaps subdivide or trim?

A-C D : It counts for me that it is in a whole because there are all the variables that I identified as reasons of the choice of IFF. As other reviewers made no remark, we have decided to let it like that.

Here add more substantive interpretation of the correlations. Give us the meaning, not just reading the table.

A-C D : The meaning of the correlation is explained in the following lines, with the interpretation of relationships with other variables depicting the same phenomenon. According to me, it is an interpretation and not only a reading.

“The choice of GBI model could be explained by this current workload, involving the need for an increase of revenue. So, the solution considered could be higher milk production and the breeding of more cows rather than diversification of activities and self-valorisation activity, the development of which requires a lot of time. Samson et al. (2016) [14] confirmed this in the Netherlands by highlighting a nearly significant effect of labour productivity on the DP increase strategy.”

Reviewer #3: The paper is clear and pleasant to read. Reporting the results of a survey on the ideal future farm for Walloon dairy farmers, this empirical research sheds light on the (ex ante) attitudes of farmers. The methods used are simple and do not add value, but they are clearly explained and used correctly. The results of the analysis are clearly explained. I would like to stress that a major effort has been made to compare the results with those of other neighbouring areas (France etc.). The comparison is of particular interest to the reader, as it allows us to situate this research work in relation to the existing literature, but also to underline the originality of the work and the new knowledge provided by this applied work on the aspirations of dairy farmers in Wallonia .

A few remarks:

- The analysis of the survey focuses on farmers' responses to questions regarding their perception of the ideal future farm. On page 13, it is written that "no opinion" farmers are excluded from the analysis to avoid potential bias. However, I would have liked to have seen a specific paragraph in which these "no opinion" farmers are analysed, even if only briefly. For example, is the answer 'no opinion' characteristic of a particular type of farmer? Furthermore, the proportion of these respondents is itself an indication of the lack of perspective that some dairy farmers may feel. The analysis of this sub-sample can therefore complement the overall analysis.

A-C D : Frequencies and means of variables for the no-opinion producers are now added. We have also performed a comparison with the complete sample to characterize this group (lines 239-258). Your second remark is really interesting. Indeed, 15% of producers may miss of perspective. I saw this conclusion in no other studies. I tried to explain this phenomenon thanks to the results observed.

- The MCA is carried out correctly, and the conclusions you draw show that there are two main types of farmers, both in terms of their current activity and their vision of the ideal future farm. It would have been interesting, and it is advisable, that you supplement your analysis with a hierarchical clustering, whether divisive or agglomerative. Such an analysis will provide a solid complement to your data processing and could corroborate and even support your conclusions.

A-C D : It was deliberately chosen to not create groups, to speak about tendency for a model or the other, producers being more or less convinced by the two extreme LBE and GBI model. To quantify the producers for a model or another, we have created classes along the dimension 2 which reveals to be a gradient between the two extreme ideal future farms. Moreover, almost 95% of the variability is explained by the two first dimensions. The first separation of the cluster analysis was between the No-opinion and the other producers, which permitted to identify and isolate them. The second separation would be along the second dimension, the creation of classes encountered this objective.

 - The main conclusions of the analysis highlight dairy farmers' vision of the future of their sector. As written in the text, the survey dates from 2015. Is it possible to compare the results with more recent statistics on the development of dairy activity in Wallonia? In other words, are there data to make a comparison between ex ante attitudes and ex post behaviours?

A-C D : It is an interesting remark. Indeed, those findings motivated us to make the survey again, five years later. Dairy producers are answering the survey at this time.

---

## [Decision Letter · Decision Letter 1]

9 Jun 2020

PONE-D-19-26278R1

How do dairy farmers wish their future farm?

PLOS ONE

Dear Dr. Dalcq,

Thank you for submitting your manuscript to PLOS ONE. I received the reports on your article. Both reviewers notice the effort you made to update your paper, but recommend a new wave of revisions. After careful consideration, we feel that it has merit but does not fully meet PLOS ONE’s publication criteria as it currently stands. Therefore, we invite you to submit a revised version of the manuscript that addresses the points raised during the review process.

Please read carefully the propositions of  reviewer #2 especially about the recommendations you did not took into account in your revision. I am afraid the reviewer is right about the interest of LCA vs MCA, as we expect a characterization of different profiles of dairy farmers.

But my proposition is that instead of LCA, you use HCPC (Hierarchical Clustering on Principal Components) which a natural and standard extension of MCA (see Arguelles et al. 2014,  Kassambara 2017 or the following technical report http://factominer.free.fr/more/HCPC_husson_josse.pdf).

We look forward to receiving your revised manuscript.

Kind regards,

Damien Rousselière, PhD

Academic Editor

PLOS ONE

Reviewers' comments:

Reviewer's Responses to Questions

**Comments to the Author**

1. If the authors have adequately addressed your comments raised in a previous round of review and you feel that this manuscript is now acceptable for publication, you may indicate that here to bypass the “Comments to the Author” section, enter your conflict of interest statement in the “Confidential to Editor” section, and submit your "Accept" recommendation.

Reviewer #2: (No Response)

Reviewer #3: All comments have been addressed

2. Is the manuscript technically sound, and do the data support the conclusions?

Reviewer #2: No

Reviewer #3: Yes

3. Has the statistical analysis been performed appropriately and rigorously? 

Reviewer #2: Yes

Reviewer #3: Yes

4. Have the authors made all data underlying the findings in their manuscript fully available?

Reviewer #2: No

Reviewer #3: Yes

5. Is the manuscript presented in an intelligible fashion and written in standard English?

Reviewer #2: No

Reviewer #3: Yes

6. Review Comments to the Author

Reviewer #2: In this revision, the authors have significantly improved the English language translation and clarified their research questions, but they have failed to address the shared concerns of reviewers regarding methods and clarity of writing. They have not adequately addressed the substance of the comments from reviewers.

Regarding methods, the reviewers still do not offer a compelling and clear justification for why MCA is appropriate for their goals and they do not offer the latent class analysis that I suggested or the hierarchical clustering (related) suggested by reviewer 3 as alternatives or robustness checks. MCA, while performed adequately, is not well suited to the way they discuss their results. They continually refer to “types” of respondents, which is what latent class analysis is for. MCA is about identifying clustering of variables, not clusters of respondents. All of their interpretation of results is about clustering and patterns of PEOPLE, not variables. This indicates a significant misalignment between the method and the research goals.

The authors do not adequately discuss the two dimensions of the MCA (figure 1) and the axes are not adequately labeled. It is not made clear why the second dimension is retained.

While the authors have provided a link to the survey online, that link is only in French and requires registration with an email address before it can be viewed, so that is not adequate. The authors still have not addressed the real concern that I raised in my previous review: they need to be clear about how they measured their concepts (intensive/extensive, etc.), what specific survey questions were used, and how those variables were coded. A clear list of questions for each concept and the coding for each is needed. For instance, in Table 2, it is not clear what specific survey questions or variables represent these concepts and how those variables were actually coded.

The authors still do not address response rate for the survey. They do now address representativeness of their respondents for this specific region, but they have not addressed the bigger questions of representativeness: how does this one region in one nation represent that nation, Europe, and/or agriculture broadly?

In all tables the n, or respondent totals, should be clear.

For Table 5 there are subscripts/footnotes that are never defined or labeled.

For Tables 3 and 4, no significance tests are reported.

The interpretation of the MCA results is circular logic. They define the clusters based on variables such as the attitude toward technology and then present a finding that people who are in the “pro-technology” GBI cluster have more positive attitudes towards technology. Of course, that is how you defined the scale in the first place.

Regarding writing, the presentations of results and its mixing with discussion of existing literature is still extremely unclear and difficult to follow. Both reviewer 1 and myself raised this critique: presenting your results intermingled with other literature is difficult to read and makes it unclear what your key findings are. This is not about the technical requirements of the journal. In its current presentation, readers cannot easily identify what your key findings are in each subsection and it is very difficult to read. For instance, in the section on pages 15-16, the authors spend substantially more time discussing other studies than they do their own results.

The introduction is improved, but still weak. The first paragraph is overly general and does nothing to build the focus of the paper. The authors also spend too much time asserting the contribution of their study before they have even reviewed the literature or told us what their analysis will be.

The attempts to incorporate new literature are cursory.

The paper is not appropriately written for a general audience. They assume too much prior knowledge from readers regarding methods both methods and the case.

Overall, the authors have inadequately addressed the careful feedback of the reviewers and made inadequate improvements. Throughout the response to reviewers they reject several important critiques with no justification of their rejection.

A number of specific points are highlighted below:

Line 79- How was that ensured (respondent producers were asked not to take into account their current farm when considering their IFF)?

Line 121- survey link not accessible without registering. Include in appendix?

Line 146- What is WARD?

Line 145- What are “particular characteristics” beyond no-opinion profiles

Line 157- What?

Line 160- What are the quantitative variables?

Table 1- What are the ns? Maybe a total figure? Percentages?

How many questions in each dimension?

Line 180- specifics of response rate still missing.

Table 2- list questions? Totals

Figure 1 define dimensions. What are the percentages in the axes labels?

Line 335- where is this figure reference from

Table 5-footnotes? Which is LBI and which GBI

What test is used here?

Line 356 explain what they mean by introduced as fixed effect

Line 620- What is SFI

Reviewer #3: Thanks to the authors, ho made signficant improvements to the paper. The full potential of the data is now revealed in the analysis. I particularly appreciate the improvements on the description of the « no-opinion » farmers, as sugested in my first review.

Specific comments :

84 : add references to « This change implied the disappearance of regulation of dairy supplies and caused volatility and decrease in the milk price »

99-100 : the question #2 is unclear. « What is the proportion of producers desiring the different IFF? » should be rephrased a bit maybe. The expression « the different IFF » will be vague for the readers. Is the IFF always different from the current farm ? And I guess « their IFF » is more thuitable thant « the IFF » as each respondent will provide a personal definition of their IFF.

122 : you do not answer to another reviewer’s comment, who wanted to know the response rate to the interview. To how many farmers was this survey submitted ? e.g. number of farmers buying the « specialised press », number of advertisements sent with the milk payment letter etc.

240:253 : the description of the no-opinion farmers adds value to the data analysis. However I am not sure that the last sentence is useful. This is your personal interpretation, but the data do not permit to reveal it.

Tables 3 and 4 : you have two columns, which are « complete sample » and « no-opinion farmers ». A third column which reflects the sample excluding the no-opinion farmers will permit the reader to compare the no-opinion ones with the others.

333 : as already said, I think this title is not well written and could be more explicit.

350 : pigeonholes. Could you be more precise ?

7. PLOS authors have the option to publish the peer review history of their article (what does this mean?). If published, this will include your full peer review and any attached files.

Reviewer #2: No

Reviewer #3: Yes: Arnaud Rault

---

## [Author Response · Author response to Decision Letter 1]

8 Jul 2020

All these elements are in the document : Response to reviewers.

Sincerely,

Anne-Catherine Dalcq

Dear Academic Editor, dear Reviewers,

First, I would like to thank you for the time spent to improve the understanding of this manuscript. This letter is organized in two parts. The first part deals with the main issue of the review (i.e., the method mobilized, which corresponds to the point raised by the academic editor and one remark of the reviewer). The second part concerns the specific answers to the specific remarks formulated by the reviewers.

Sincerely,

Anne-Catherine Dalcq 

1/ Methodology

Please read carefully the propositions of reviewer #2 especially about the recommendations you did not took into account in your revision. I am afraid the reviewer is right about the interest of LCA vs MCA, as we expect a characterization of different profiles of dairy farmers.

But my proposition is that instead of LCA, you use HCPC (Hierarchical Clustering on Principal Components) which a natural and standard extension of MCA (see Arguelles et al. 2014, Kassambara 2017 or the following technical report http://factominer.free.fr/more/HCPC_husson_josse.pdf).

H. Soyeurt : 

Dear Editor,

Dear Reviewers,

Instead of my PhD student, Miss Anne-Catherine Dalcq, I would like to answer to the question related to the methodology used in this study. I am Professor Hélène Soyeurt and I teach courses related to Data Mining and Machine Learning at Gembloux Agro-Bio Tech (University of Liège). Therefore, I have an experience in the use of multivariate analysis. It is why I would like to answer by myself to the question related to the method used in this article. 

Before starting an explanation, I would like to precise that we have not really understood the comments of the reviewer about Latent Class Analysis (LCA) because the objective was not to create groups of farmers. Indeed, and this is the innovative aspect of this paper, a gradient between two quite different models of farms was studied. The gradient is really important because it appears to us simplistic to classify farmers in only 2 groups, working with a gradient allowed us to nuance the position of the dairy producers and to analyze more precisely the link between this position and other characteristics. The use of the gradient allows studying the trend of a farmer. It is why in this paper, we always mention “tend towards” to make a reference to the position on the gradient and not a binary choice of a model. 

As we did not want to create farmer’s clusters, LCA was not appropriate as well as HCPC. However, as asked by the second reviewer to prove the robustness of our approach, we have decided to show you the similarities and the extended work that it is possible to do using Multiple Correspondence Analysis (MCA) and LCA. Indeed, it is also possible to use LCA to create a gradient instead of using the clusters. 

So, I would like to remind the methodology that we have proposed in this study (see the figure below, Fig 1)). Again, MCA was used to observe the relationships between the seven studied variables. Based on the interpretation of MCA dimensions, we have observed that the second dimension represented the positioning between the two models for a dairy farm. Therefore, the score for this dimension for a specific producer allows to know its perception of ideal future farm between the two extreme models. This gradient allows avoiding to limit the farm typology to 2 clusters. This is interesting because many farmers combined some approaches specific to one model of farms (no more intensification but extensification, local-based market) or another one (global-based market, continuously improvement of the productivity thanks to, notably, intensification, …). Therefore, wishing to split the dairy farmers into 2 groups is too limitative. Some farmers remain between these two models (Fig 2), some farmers are more convinced than others by a model. Farmers showing higher scores are really convinced by GBI model, farmers showing lower but positive scores choose GBI model but are not 100% in this way of farm development. Negatives scores express the position of farms in favor of LBE model, the lowest scores reflected pronounced adhesion to this model. It is why the gradient as proposed in this paper was useful and seem to us the most interesting tool to represent dairy producers.

Figure 1 Methodology

Figure 2 Distribution of the producers (the "with an opinion" ones) along the second dimension of the MCA

Similarly, we have done this job using LCA methodology as requested by the reviewer#2. LCA allows defining clusters from the dataset. In this case and to be in line with the study objective, we have decided to create 3 clusters. Moreover, based on the AIC and BIC values, it was also the best model (Fig 3). LCA allows us to create those 3 clusters. After the interpretation of defined clusters, it appeared that we have a cluster representing “No opinion” responders, one cluster representing “global-based intensive” (GBI) farmers and one cluster representing “local-based extensive” (LBE) farmers (Fig 4). Therefore, those clusters were similar to the ones obtained by combining MCA and Ward clustering (=HCPC method) as done to clean the dataset (i.e., extract responders with “no opinion” behavior). Again, we did not want to use those clusters to make our analysis. But, using this methodology, it is possible to obtain a probability to belong to a specific cluster. Therefore, we have decided to compare the gradient as defined in the current study to the probability to belong to the LBE or GBI clusters defined using the LCA method. 

So, now, it is time to present you the results. I will not present you the results about MCA as those results are reported in the article. In this paragraph, we will focus on the LCA results. As we used categorical variables which are not ordinal, we have decided to use the polytomous latent class analysis. The variables used, called manifest variables, were the same than the one used for MCA. The modalities for each variable were recoded from 1 to 3:

 Intensification: 1. Intensive, 2. Extensive, 3. No opinion

 Number of activities: 1. Specialised, 2. Diversified, 3. No opinion

 Technology: 1. Strongly based on new technologies, 2. Weakly based on new technologies, 3. No opinion

 Workforce: 1. Familial workforce, 2. Employed workforce, 3. No opinion

 Kind of management: 1. Group of farmers management, 2. Independent farmer management, 3. No opinion

 Market: 1. Global market, 2. Local market, 3. No opinion

 Milk quality: 1. Standard quality milk, 2. Differentiated quality milk, 3. No opinion

As this clustering is very sensitive to the prior values used to start the iteration, 10 repetitions were used to provide you the final results for all models. All calculations were done with R software and more specifically the package poLCA. First, we have run different LCA models using a different number of classes (clusters). We have tested models from 1 to 10 classes and then we have estimated the AIC and BIC criteria to observe which model allowed the best fitting. The results of AIC and BIC for all models are presented in the figure below (Fig 3). From this figure, we can conclude that the model allowing the creation of 3 classes is a good comprise between BIC and AIC (i.e., the lowest BIC and AIC values). 

Figure 3 AIC & BIC criteria values

The figure below shows you the clusters defining by the model allowing a discrimination of the data into 3 clusters (Fig 4). This figure represents the probability to have a specific modality for each cluster. So, based on those results, we can conclude that the first clustering (class 1) is related to GBI dairy producers, the second cluster (class 2) is related to LBE dairy producers and the last cluster (class 3) is related to the dairy producers with “no opinion”.

Figure 4Description of the three clusters obtained by LCA method

Therefore, in order to clean the dataset (i.e., delete records from farmers having many “no opinion” views), we can use the clustering 3. This process is similar to the MCA + Ward clustering (HCPC) proposed in the manuscript. Again, we did not want to use clusters but we want to use a gradient. For MCA, this gradient was the second MCA dimension. In the context of LCA, this gradient can be derived from the probability to belong to class 1 (“GBI producer”) or class 2 (“LBE producer”). So, to show the robustness of the MCA approach used, we calculated the correlation between those probabilities and the score of the second MCA dimension (called gradient in the manuscript). The correlation between the probability to belong to class 1 and the gradient was equal to 0.83. The correlation between the probability to belong to class 2 and the intensification gradient was equal to -0.87. The correlation between the 2 probabilities was equal to -0.97.

From all of those results, you can see that the relationship is strong between MCA and LCA using an innovative approach focusing on the score/probability of an individual and not directly to a cluster. Using only the second dimension of MCA, we can reflect both clusters (class 1 and class 2) simultaneously. Indeed, the second dimension is a gradient “GBI-LBE” and included both. This is really interesting to observe the relationships between this gradient and other quantitative or qualitative variables as now the studied trait is quantitative. It allows to see if a modality of a categorical variable is the choice of really convinced GBI dairy producers or dairy producers only sticking out of the GBI model.

I hope that this demonstration illustrates well the relevancy of the approach proposed in our paper. A sentence will be added in the materials and methods section to explain why LCA was not used. Moreover, some articles exist also in the literature to prove the mathematical relationships between MCA and LCA (e.g., Lautsch and Plichta, Psychology Science 2003:298-323 as well as Van der Heijden, et al., Sociological methodology 1999).

The fact that the two dimensions explained almost 95% of the variability and that all the modalities representing an opinion positioning themselves along the second dimension support us to take the second dimension as gradient, after deletion of the No-opinion producers which were discriminated by the first dimension. 

Finally, I would like to acknowledge you for the great job done in the review of this paper. The manuscript was improved a lot. 

Sincerely,

Prof Hélène Soyeurt

PS : The answers to all other comments were done by Anne-Catherine Dalcq. 

2/ Others comments

Reviewer #2: In this revision, the authors have significantly improved the English language translation and clarified their research questions, but they have failed to address the shared concerns of reviewers regarding methods and clarity of writing. They have not adequately addressed the substance of the comments from reviewers.

Regarding methods, the reviewers still do not offer a compelling and clear justification for why MCA is appropriate for their goals and they do not offer the latent class analysis that I suggested or the hierarchical clustering (related) suggested by reviewer 3 as alternatives or robustness checks. MCA, while performed adequately, is not well suited to the way they discuss their results. They continually refer to “types” of respondents, which is what latent class analysis is for. MCA is about identifying clustering of variables, not clusters of respondents. All of their interpretation of results is about clustering and patterns of PEOPLE, not variables. This indicates a significant misalignment between the method and the research goals.

A-C Dalcq: First, we would like to thank you for the deep reading done on this article and for your formulated comments. We missed your request to test the robustness of our method with other ones. We apologize for this mistake. You will find our work of comparison with LCA method in the first part of this letter. In the article, we speak about producers tending towards “GBI” or “LBE” models to express the results coming from the use of our gradient. Indeed, MCA is a method to identify relationships between variables but is also suitable to make groups of individuals thanks to the use of its extension HCPC, as proposed by the academic editor. A detailed information about the comparison of MCA and LCA is now given in the first part of this letter. 

The authors do not adequately discuss the two dimensions of the MCA (figure 1) and the axes are not adequately labeled. It is not made clear why the second dimension is retained.

 A-C Dalcq: We added some information. The updated explanations of the two dimensions are taken up just after (Lines 254-261/286-306/339-343). Do you need more information? If it is the case, could you guide us?

Remark : Lines specified throughout this letter are those of the revised manuscript with track changes.

The axes are labeled as the figure is provided by SAS 9.4. We precised its meaning thanks to the caption and the following interpretation.

Explanations of the two dimensions:

Lines 254-261: “The first dimension of MCA showed positive relationships with the modalities no opinion of each characteristic and negative relationships with all the modalities representing an opinion. Thus, the first dimension of the MCA allowed permitted differentiation between the producers who did not give their opinion concerning characteristics of IFF and the producers who did (Fig 1). Cluster analysis was used to isolate the group of producers with a lot of ‘no opinion’ answers to the seven questions: this formed the first separation of classes of the analysis, dividing the “no-opinion” producers (15%) from the others (85%).”

Lines 286-306: “The second dimension of the MCA showed positive relationships with some modalities of the IFF characteristic and negative relationships with their opposite. Thus, this dimension seems was the most interesting for highlighting to highlight the wishes of dairy farmers about their IFF, for those who took a position on this question. More precisely, this axis showed a gradation of question modalities and proximity between several characteristics. This dimension led to the identification of two extreme tendencies (Fig 1); the modalities of familial workforce, independent farmer management and management by a group of farmers were near to zero on this axis (Fig 1). This means that the small proportion of producers supporting group management was distributed between the two extreme tendencies observed. The position of the modalities of familial workforce and independent farmer at the middle of the second dimension illustrated the fact that these modalities were chosen by producers from the two tendencies identified. The small proportion of producers choosing an employed workforce was positioned at the top of the second dimension (Fig 1).”

“The first tendency, related to high scores on the second MCA dimension, corresponds to IFF with the following characteristics: global market, standard milk, intensive system, employed workforce, specialised and strongly based on new technologies.”

Lines 339-343: “The second tendency, contrary to the first tendency, was characterised by negative scores on the second MCA dimension. This axis was represented by the following modalities: weakly based on new technologies, diversified, differentiated quality milk, local market and extensive system (Fig 1). This reflects another form of dairy farming. »

Reason of the use of the second dimension: 

Lines 379-384: “To study the relationships between the different IFF, the reasons for these and other interesting technico-economic information, the second dimension was considered as a gradient (IFFg) interpreted at the extremities as global-based intensive producers (GBI: high positive scores) and local-based extensive producers (LBE: high negative scores). The choice to work with a gradient rather than a clear separation of the two tendencies was motivated by the will to not put dairy producers into boxes pigeonholes”

While the authors have provided a link to the survey online, that link is only in French and requires registration with an email address before it can be viewed, so that is not adequate. The authors still have not addressed the real concern that I raised in my previous review: they need to be clear about how they measured their concepts (intensive/extensive, etc.), what specific survey questions were used, and how those variables were coded. A clear list of questions for each concept and the coding for each is needed. For instance, in Table 2, it is not clear what specific survey questions or variables represent these concepts and how those variables were actually coded.

A-C Dalcq: In the current version, we have added an annex (Appendix 1) with the translation of the questions mobilized in the paper. Given the length of the survey, we provided only the questions raised in the present paper. Some information are mentioned at lines 129-136 about the questions related to the ideal future farm characteristics (intensive vs. extensive,…), the way of measurement for our developed concepts. If you need more information, could you precise them to us explicitly ?

Lines 129-136: “The entire survey was composed of 127 questions where the answers were decomposed into 498 categorical and 44 quantitative variables. The question ‘Without taking into account your current farm, what is, according to you, the ideal future farm to ensure a revenue?” was proposed to the producers and they must could choose between short propositions on seven items: 1) intensive or extensive production; 2) specialised or. diversified activity (or activities); 3) farming strongly or weakly based on new technologies; 4) farm managed by an independent farmer or a group of managers; 5) family or employed workforce; 6) providing production for local or global markets; 7)providing standard or differentiated quality production. The modality “no opinion” was available for each IFF question.”

The authors still do not address response rate for the survey. They do now address representativeness of their respondents for this specific region, but they have not addressed the bigger questions of representativeness: how does this one region in one nation represent that nation, Europe, and/or agriculture broadly?

A-C Dalcq: The response rate of 6,1% was already precised in the past manuscript. You can find it at lines 200-201 of the current text. 

Lines 200-201 : “The sample set of 245 producers represented 6.1% of the dairy producers in Wallonia (about 4,000 dairy producers in 2015 and 3,500 in 2017 (STATBEL, 2019)).”

For the second part of your comment, do you want that we precise that the Walloon Region is one of the two regions of Belgium, which is one of the 27 members countries of the European Union? If yes, the proposal could be : “The Walloon Region is one of the two regions of Belgium, which is one of the 27 members countries of the European Union”. 

Or do you want that we precise the number of dairy producers in Belgium and in the European Union? The International Dairy Federation mentions 9,674 dairy farms in Belgium and 1,130,700 farms with a dairy activity in the European Union (Confédération Belge de l’Industrie Laitière, 2020). The number of Walloon dairy farms is obviously low amongst all of these countries. We do not know if all this information is relevant to be written in the manuscript. 

Complete reference: Confédération Belge de l’Industrie Laitière. 2020. Rapport Annuel 2020.

The goal of this paper is to inform about the position of dairy producers of a region, which is moreover quite heterogeneous regarding the geopedologic conditions (lines 205-206), this one can represent the context and the resources of other producers in Europe. The goal was not to give a complete vision of all the European producers, which needs higher means.

In all tables the n, or respondent totals, should be clear.

A-C Dalcq: We added a sentence at lines 395-396 :“These analysis were conducted on the producers who have an opinion (N = 207).” And the N was precised and added in each table. Thanks for this remark which brings clarity throughout all the paper.

For Table 5 there are subscripts/footnotes that are never defined or labeled.

A-C Dalcq: The subscripts are now precised in each table. “Means with different letters are significantly different.”. Thanks for this remark.

For Tables 3 and 4, no significance tests are reported.

A-C Dalcq: Indeed, the goal was not to test the differences between the no-opinion producers and all the sample but to give an idea of the characteristics of the No-opinion producers. Reviewer#3 asked us to give all of this information also for producers with an opinion. We realized ANOVA tests between the no-opinion producers and the producers with an opinion (lines 280-284)

The interpretation of the MCA results is circular logic. They define the clusters based on variables such as the attitude toward technology and then present a finding that people who are in the “pro-technology” GBI cluster have more positive attitudes towards technology. Of course, that is how you defined the scale in the first place.

A-C Dalcq: The variable “technology” is “Mechanisation and robotisation : help for workload and administrative aspects “ and is present in the part “Reasons”. This result is presented to explain one reason of the producers tending towards “GBI-model” to tend to this model and one of its component, the technology. We have better precised our idea by adding the sentence: “We observed that the wish of technology of producers tending towards GBI model can be explained by the fact that they considered it as help for workload.” at lines 534-536. 

Regarding writing, the presentations of results and its mixing with discussion of existing literature is still extremely unclear and difficult to follow. Both reviewer 1 and myself raised this critique: presenting your results intermingled with other literature is difficult to read and makes it unclear what your key findings are. This is not about the technical requirements of the journal. In its current presentation, readers cannot easily identify what your key findings are in each subsection and it is very difficult to read. For instance, in the section on pages 15-16, the authors spend substantially more time discussing other studies than they do their own results.

A-C Dalcq: As already mentioned and visible at lines 304-364 in the revised paper, the explanations based on the findings of other past studies help to explain the relationships observed between the modalities of the seven ideal future farm characteristics. 

The introduction is improved, but still weak. The first paragraph is overly general and does nothing to build the focus of the paper. The authors also spend too much time asserting the contribution of their study before they have even reviewed the literature or told us what their analysis will be.

A-C Dalcq: What could be your expectations about the structure of this introduction? As the reviewer#3 did not make comments about the redaction of this part and without deeper expectations from your, the structure of the introduction was not changed.

The attempts to incorporate new literature are cursory.

A-C Dalcq: The work of Mr Mooney was consulted and two references was added to the paper (lines 236-238, lines 577-580). Did you expect references to more elements of his work? Could you precise which ones? Moreover, we also investigated the phenomenon of bifurcation. Literature about bifurcation was mainly found for the organic activity. We consulted a Professor with skills in sociology, Prof Kevin Maréchal, of Gembloux Agro-Bio Tech-University of Liège (Belgium), who provided us also this literature reference. Do you have other literature to advice to me? 

The paper is not appropriately written for a general audience. They assume too much prior knowledge from readers regarding methods both methods and the case.

A-C Dalcq: We have now added information about the choice of the method between MCA and LCA, and about the method WARD (regarding a following remark). We hope that it brings the missing information. If this information is not complete for you, could you precise us explicitly the requested information?

Lines 165-170: “This method was chosen instead of the creation of classes, possible with the Latent Class Analysis method or the Numerical Classification on the scores of MCA (Hierarchical Clustering on Principal Components). This choice was motivated by the wish to not put the producers in boxes but study their position on a gradient between potential extreme models identified along the dimension.”

Lines 153-156 : “The WARD method is a hierarchical agglomerative method (Everitt et al., 2011). The principle of this kind of method is to put initially the n individuals in n groups and then to agglomerate the groups. The algorithm of WARD makes it in such a way that the gatherings induce the lowest decrease of R2 at each step.”

Overall, the authors have inadequately addressed the careful feedback of the reviewers and made inadequate improvements. Throughout the response to reviewers they reject several important critiques with no justification of their rejection.

A-C Dalcq: We recognized that we do not explain the choice of the method and we do not test its robustness with statistic treatments. We apologized for that. We missed this request. We realized this analysis at this time.

A number of specific points are highlighted below:

Line 79- How was that ensured (respondent producers were asked not to take into account their current farm when considering their IFF)?

A-C Dalcq: It is now precised in the Materials and methods section (Lines 131-134). Thanks for your remark. 

Lines 131-134 : The question ‘Without taking into account your current farm, what is, according to you, the ideal future farm to ensure a revenue?” was proposed to the producers and they must choose between short propositions on seven items: 1) intensive or extensive production; 2)[…]”

Line 121- survey link not accessible without registering. Include in appendix?

A-C Dalcq: The appendix is realized and available in the new submission.

Line 146- What is WARD?

A-C Dalcq: We added sentences of explanation about the Ward method at lines 153-156. It is a hierarchical agglomerative method of Numerical classification. 

Lines 153-156: “The WARD method is a hierarchical agglomerative method (19). The principle of this kind of method is to put initially the n individuals in n groups and then to agglomerate the groups. The algorithm of WARD makes it in such a way that the gatherings induce the lowest decrease of R2 at each step.”

Line 145- What are “particular characteristics” beyond no-opinion profiles?

A-C Dalcq: We replaced particular by “some” (Lines 151). We hope it makes it clearer.

Line 157- What?

A-C Dalcq: This sentence is a part of the method. As explained and precised before, we realized a MCA on the seven ideal future farm questions. We observed that the modalities “no-opinion” of the seven questions gathered and were positively related with the first dimension of the MCA. All the modalities reflecting an opinion were negatively related to the first dimension. Thus, the first dimension allowed to differentiate the producers with an opinion or not. 

The modalities “intensive”, “global-market”, “specialized”, “standard quality milk”, “employed workforce” and “strongly based on new technologies” gathered and were positively related to the second dimension. The modalities “extensive”, “local-market”, “diversified”, “quality differentiated milk” and “lowly based on new technologies” gathered and were negatively related to the second dimension. The second dimension appeared to us as a gradient between the two extreme models of ideal future farm “Global-based intensive” and “Local-based extensive”. 

The two dimensions explained almost 95% of the variability of the dataset. Therefore, the study of only these two dimensions appeared to us relevant. Then, we realized a numerical classification on the scores on the two dimensions of the MCA (statistic treatment equivalent to HCPC- Hierarchical Clustering on Principal Components). The first two groups created were the “no-opinion” producers and the producers “with an opinion”. This allowed us to exclude the “no-opinion” producers and to study the producers with an opinion thanks to the second dimension, this one had at its extremities the “Local-based extensive model” and the “Global-based intensive model”. But the producers with an opinion distributed themselves along this dimension. Thus, we decide to work with the second dimension, as a gradient of ideal future farm (Fig 5). 

Then we want to study the relationships between this ideal future farm gradient and the other information present in the survey.

The relationships between the ideal future farm gradient and the categorical variables of the survey were studied thanks to generalized linear models.

The gradient was the y, the variable to explain. The modalities of the categorical variables were the fixed effect of the generalized linear model, the factors explaining.

y = effect + residual

Where y was a vector contained the score on the ideal future farm gradient (the second dimension of MCA); effect was the qualitative variables of the survey. In other words, the model was :

Ideal future farm gradient = categorical variable + e

To study the relationships between the gradient and the quantitative variables of the survey, correlation coefficients and their level of significance were calculated.

Line 160- What are the quantitative variables? 

A-C Dalcq: Quantitative is a statistical term defining a continuous numerical variable. Do you want that we use the term « numerical » ? But this term is less precise as it does not reflect the continuous dimension of the variable. 

Table 1- What are the ns? Maybe a total figure? Percentages? 

A-C Dalcq: Absolute frequencies named counts. We precised at lines 186 and 194: “Absolute frequencies (counts)”.

How many questions in each dimension? 

A-C Dalcq: One question. We have now mentioned that in the Materials and methods section at Lines 187-188: “and of the answer to the question which corresponds to this corresponding characteristic for the current situation”.

Line 180- specifics of response rate still missing. 

A-C Dalcq: The response rate is 6,1% (Line 200). Which supplementary indication do you need? We gave you numbers of farms with a dairy activity in Belgium and in European Union in this letter. We give also more information about the conditions where the survey was communicated to the producers to give you an idea of the way of proceeding. More information is given at lines 120-126.

Lines 120-126: “We communicated with Walloon dairy producers about the goals of the survey and its access broadly via all communication ways towards them : specialised press, agricultural internet websites, Unions and also advertisements through the milk payment letter which is sent to all the Walloon dairy producers once a month. The survey written in French can be viewed at the following internet link: https://www.gembloux.ulg.ac.be/enquete/index.php/219425?lang=fr and its English translation is viewable in the Appendix“

Table 2- list questions? 

We have added the question in the table (Line 223).

Question

Without taking into account your current farm, what is, according to you, the ideal future farm to ensure a revenue?” Proposition Percentage (%)

Intensive vs. extensive Intensive 43

 Extensive 30

 No opinion 27

Specialised vs. diversified Specialised 43

 Diversified 47

 No opinion 10

Strongly vs. weakly based on new technologies Strongly 35

 Weakly 41

 No opinion 24

Managed by an independent farmer vs. a group of managers Independent farmer 72

 Group of managers 18

 No opinion 10

Family vs. employed workforce Family 87

 Employed 5

 No opinion 8

Providing dairy production for local vs. global market Global 43

 Local 32

 No opinion 25

Providing standard vs. differentiated quality dairy production Standard 38

 Differentiated quality 45

 No opinion 17

Totals 

A-C Dalcq: It is now mentioned at line 223.

Figure 1 define dimensions. What are the percentages in the axes labels? 

A-C Dalcq: The percentages in the axes labels are the inertia but these ones underestimated the part of information explained by the dimensions. The corrected inertia values were calculated. This is explained in the Materials and methods section at lines 142-146: “For a MCA, the eigenvalue of the dimensions generated, named principal inertia, is a biased measure of the amount of information presented by a dimension (Palm, 2007). Corrected inertia rates were calculated, as described by Benzécri (Benzécri, 1979), to quantify the correct proportion of information of a dimension.”

The corrected values of inertia are presented at lines 248-250. The nature of the values are precised in the caption of the figure (lines 251-253).

“The percentage of principal inertia of the dimensions 1 and 2 of MCA were 16.75% and 12.38%, respectively (Fig 1). The value of corrected inertia for the two first dimensions reached 72.7% and 21.5% respectively, gathering almost 95% of the information. 

Fig 1. Representation of the modalities in the multiple correspondence analysis first factorial plan. Values of principal inertia reached 16.75% and 12.38%. Values of corrected inertia reached 72.7% and 21.5%.”

The correction of Benzécri is made following this calculation: 

corrected inertia=〖(s/(s-1))〗^2*(µ_k-1/s)^2 with µ_k> 1/s

s = number of categorical variables in the MCA

µk= eigenvalue of the dimension

(0.33508 and 0.2475, eigenvalues of respectively the first and second dimension of the present MCA) 

The corrected inertia gives a better appreciation of the amount of information explained by each dimension (Benzécri, 1979), than the inertia automatically provided by the software.

We did not think necessary to precise this in the paper but we provide the reference where this calculation is presented.

Line 335- where is this figure reference from 

A-C Dalcq : We did not provide a representation of the distribution of the producers along the second dimension, we did not think it was necessary as we provided the percentages and this figure would lengthen the paper. But following your request, we have now added this figure in the present manuscript (Lines 376-377).

Fig 2. Distribution of the producers along the second dimension (the dotted line represents the mean score on the second dimension of the producers)(N = 207)

Table 5-footnotes? Which is LBI and which GBI 

A-C Dalcq : We were not sure about the goal of your question. We precised the function of the letters a and b. Concerning your request about the LBE and GBI, as explained with more details in the point 1/, the analysis is not done between groups LBE or GBI and the variables present in the survey but between an ideal future farm gradient and the variables present in the survey. The means presented in the tables 5, 7 and 8 are the mean value of this gradient for the different modalities of the categorical variables. We explained this at lines 390-395.

Lines 390-395: “Tables 5, 7 and 8 give the results of generalised linear models where the categorical variables were introduced separately as a fixed effect in the model. Significantly lower estimates of IFFg for a specific modality of the considered categorical variable depicts a tendency of producers desiring a LBE model to choose this modality, while significantly higher estimates of IFFg means a tendency of producers wanting a GBI model to choose this modality.” 

What test is used here? 

A-C Dalcq: Generalised linear models were used to study the level of significance of the differences between the means values of the gradient of the modalities of categorical variables: y was the ideal future farm gradient and the effect included in the model was the categorical variable. We explained this part of the method in the Materials and methods section at lines 178-184.

Lines 178-184: “For categorical variables, the scores of MCA dimensions were modelled using these variables as a fixed effect in a generalised linear model. Least squares means were estimated for the two-by-two comparisons using the Tukey test. The level of significance of those differences was assessed based on the P-value of the test. For quantitative variables, Pearson correlation coefficients were calculated between the scores of MCA dimensions and these variables. Their corresponding P-values were estimated to observe if the correlation values were significantly different from 0.”

Line 356 explain what they mean by introduced as fixed effect 

A-C Dalcq: As explained in the point 1/ of this letter, the gradient was used as y (variable to be explained) of the generalized linear models and the fixed effect introduced in the generalized linear model was the categorical variables. See the previous answer related to the same topic for more details.

Line 620- What is SFI 

A-C Dalcq : We precised this at line 661. “SFI = study, formation and information ». It is after the table. We have now added an asterisk to highlight the explanation of this abbreviation.

Reviewer #3: Thanks to the authors, ho made significant improvements to the paper. The full potential of the data is now revealed in the analysis. I particularly appreciate the improvements on the description of the « no-opinion » farmers, as suggested in my first review.

A-C Dalcq: Thank you for the interest given to this study.

Specific comments:

84: add references to « This change implied the disappearance of regulation of dairy supplies and caused volatility and decrease in the milk price »

A-C Dalcq: We added a reference and precised our purpose (Lines 83-84). Thanks for your remark.

Lines 83-84: “This change implied the disappearance of regulation of dairy supplies and was bringing uncertainty about the milk price (16). caused volatility and decrease in the milk price. “

Reference 16: Salou, T., H.M.G. van der Werf, F. Levert, A. Forslund, J. Hercule, and C. Le Mouël. 2017. Could EU dairy quota removal favour some dairy production systems over others? The case of French dairy production systems. Agric. Syst. 153:1–10. doi:10.1016/j.agsy.2017.01.004.

99-100: the question #2 is unclear. « What is the proportion of producers desiring the different IFF? » should be rephrased a bit maybe. The expression « the different IFF » will be vague for the readers. Is the IFF always different from the current farm ? And I guess « their IFF » is more thuitable thant « the IFF » as each respondent will provide a personal definition of their IFF.

A-C Dalcq: We understand your will to make this question clearer. We have replaced it by « How the dairy producers distribute themselves between IFF highlighted ?” (Lines 100-101, lines 365-366).

122: you do not answer to another reviewer’s comment, who wanted to know the response rate to the interview. To how many farmers was this survey submitted ? e.g. number of farmers buying the « specialised press », number of advertisements sent with the milk payment letter etc.

A-C Dalcq: We precised in the text that all the Walloon dairy producers received this payment letter: +/- 4,000 producers. Therefore, the 245 respondents correspond to 6,1% of the population, i.e. the response rate. We added information at lines 120-126 & 200-201. We do not know if the information number of farmers buying the « specialised press », number of advertisements sent with the milk payment letter are necessary in this paper but we added information to give an idea of the conditions in which the survey was communicated.

Lines 120-126: “We communicated with Walloon dairy producers about the goals of the survey and its access broadly via all communication ways towards them : specialised press, agricultural internet websites, Unions and also advertisements through the milk payment letter which is sent to all the Walloon dairy producers once a month. The survey written in French can be viewed at the following internet link: https://www.gembloux.ulg.ac.be/enquete/index.php/219425?lang=fr and its English translation is viewable in the Appendix.

A total of 245 producers completed our survey between November 2014 and January 2015.”

Lines 200-201: “The sample set of 245 producers represented 6.1% of the dairy producers in Wallonia (about 4,000 dairy producers in 2015 and 3,500 in 2017 (STATBEL, 2019).”

240:253 : the description of the no-opinion farmers adds value to the data analysis. However I am not sure that the last sentence is useful. This is your personal interpretation, but the data do not permit to reveal it.

A-C Dalcq: Indeed, this sentence was deleted (Line 279). Thanks for your remark.

Tables 3 and 4: you have two columns, which are « complete sample » and « no-opinion farmers ». A third column which reflects the sample excluding the no-opinion farmers will permit the reader to compare the no-opinion ones with the others.

A-C Dalcq: Indeed, this column was added (Lines 280, 283). And, as asked by reviewer#2, we realized (1) generalized linear models to compare the means of the quantitative variables between the no-opinion producers and producers with an opinion and (2) tests of proportion to compare the proportion of each modalities of the categorical variables between the no-opinion producers and producers with an opinion. We added description of these statistical treatments in the Materials and Methods section (Lines 157-160).

333 : as already said, I think this title is not well written and could be more explicit.

A-C Dalcq: It was changed at lines 365-366. See also the answer to your previous comment related to the same topic.

350 : pigeonholes. Could you be more precise?

A-C Dalcq: As it caused doubt in your comprehension of this idea, we replaced by boxes (Line 384). 

 Put people (here dairy producers) in boxes. 

It was used to explain the fact to put a label of someone. But in this paper, we want to nuance the position of the producer regarding its ideal future farm.

---

## [Decision Letter · Decision Letter 2]

1 Oct 2020

PONE-D-19-26278R2

How do dairy farmers wish their future farm?

PLOS ONE

Dear Dr. Dalcq,

Dear Colleague

We’re pleased to inform you that your manuscript has been judged scientifically suitable for publication. Thanks again for all the work done for this new version.

The reviewers have just some last suggestions (title, abstract, discussion, conclusion)  to improve the writing of the paper and “make it more impactful” (see  reviewer 1 comments).

Please take them into account and send us an updated paper. I will check it directly without a new round of reviews. The paper will be formally accepted for publication once it meets these last recommendations. After acceptance, some last technical requirements may be asked.

A marked-up copy of your manuscript that highlights changes made to the original version. You should upload this as a separate file labeled 'Revised Manuscript with Track Changes'.An unmarked version of your revised paper without tracked changes. You should upload this as a separate file labeled 'Manuscript'.

We look forward to receiving your revised manuscript.

Kind regards,

Damien Rousselière, PhD

Academic Editor

PLOS ONE

Reviewers' comments:

Reviewer's Responses to Questions

**Comments to the Author**

1. If the authors have adequately addressed your comments raised in a previous round of review and you feel that this manuscript is now acceptable for publication, you may indicate that here to bypass the “Comments to the Author” section, enter your conflict of interest statement in the “Confidential to Editor” section, and submit your "Accept" recommendation.

Reviewer #2: (No Response)

Reviewer #3: All comments have been addressed

2. Is the manuscript technically sound, and do the data support the conclusions?

Reviewer #2: Yes

Reviewer #3: Yes

3. Has the statistical analysis been performed appropriately and rigorously? 

Reviewer #2: Yes

Reviewer #3: Yes

4. Have the authors made all data underlying the findings in their manuscript fully available?

Reviewer #2: No

Reviewer #3: Yes

5. Is the manuscript presented in an intelligible fashion and written in standard English?

Reviewer #2: Yes

Reviewer #3: Yes

6. Review Comments to the Author

Reviewer #2: In this revision, the authors have significantly improved the manuscript and clarified their methodological choices. There are still clarifications that need to be made to the writing.

Regarding methods, the authors have clarified their choice of MCA and their figures and tables are now appropriately labeled and described. The survey appendix is a crucial addition as it allows readers to directly see the way that the concepts were measured and to connect the results to the measurement.

In the conclusion, the authors should still address more directly how findings from this region hold lessons and relevance for other regions.

Regarding writing, the presentations of results and its mixing with discussion of existing literature is still unclear and difficult to follow. I continue to advocate for separating the literature into a discussion section, following the findings. As it is currently written, it is difficult to identify what are the key findings because they are intermingled with the literature review.

The first paragraph of the introduction is still overly general and does nothing to build the focus of the paper.

A number of specific points are highlighted in the PDF.

Overall, the authors have addressed my major concern regarding methodology. Their decisions are now given appropriate explanation and the conclusions are clearly justified by the findings. The writing could still be improved to make the paper more impactful.

Reviewer #3: Thanks to the authors for proposing this reviewed version of the article "How do dairy farmers wish their future farm?".

I have had a creful attention at reading this version, and I really appreciate the efforts made to make the manuscript easier to read and understand. I do have nothing to add to my previous comments as the authors made the necessary corrections and improvements to the manuscript.

I underline the effort to answer to the recommendations of reviewer 2 on the use of LCA vs MCA. The answer is clear and exhaustive. I will not comment anymore on this and I let reviewer 2 make his own comments on that.

I only have one major recommendation. You should change the title of the article. As recommended by numbers of editors, the title should be declarative rather than a question. The title is a general question now, but your contribution is in how you address it, so you should focus on the main result, and be very precise on the content. For example, you can target your title by highlithing the different trajectories or perspective of farm evolution, depending on LBE and GBI profiles. Moreover, as it is now, the title is misleading as we do not understand that the article explores the result of an interview in Waloony. To be clear, your title should reveal your main novel finding.

As a consequence, the abstract should develop and explain your approach, leading to the restult, and then the introduction will explain in detail your contribution to knowledge and science, and how your research answers, at least in part, the question "how do dairy farmers wish their future farm".

7. PLOS authors have the option to publish the peer review history of their article (what does this mean?). If published, this will include your full peer review and any attached files.

Reviewer #2: No

Reviewer #3: **Yes: **Arnaud Rault

---

## [Author Response · Author response to Decision Letter 2]

16 Nov 2020

Thank you for your remarks. It enhanced the quality of our paper.

Sincerely,

Anne-Catherine Dalcq

---

## [Editor Report · Decision Letter 3]

19 Nov 2020

The Walloon farmers position differently their ideal dairy production system between a global-based intensive and a local-based extensive model of farm.

PONE-D-19-26278R3

Dear Dr. Dalcq,

We’re pleased to inform you that your manuscript has been judged scientifically suitable for publication and will be formally accepted for publication once it meets all outstanding technical requirements.

Kind regards,

Damien Rousselière, PhD

Academic Editor

PLOS ONE

---

## [Editor Report · Acceptance letter]

23 Nov 2020

PONE-D-19-26278R3 

The Walloon farmers position differently their ideal dairy production system between a global-based intensive and a local-based extensive model of farm. 

Dear Dr. Dalcq:

I'm pleased to inform you that your manuscript has been deemed suitable for publication in PLOS ONE. Congratulations! Your manuscript is now with our production department. 

Kind regards, 

on behalf of

Dr. Damien Rousselière 

Academic Editor

PLOS ONE